# GLOBALTOMO: A GLOBAL DATASET FOR PHYSICS-ML SEISMIC WAVEFIELD MODELING AND FWI

## ABSTRACT

Global seismic tomography, taking advantage of seismic waves from natural earthquakes, provides essential insights into the earth's internal dynamics. Advanced Full-Waveform Inversion (FWI) techniques, whose aim is to meticulously interpret every detail in seismograms, confront formidable computational demands in forward modeling and adjoint simulations on a global scale. Recent advancements in Machine Learning (ML) offer a transformative potential for accelerating the computational efficiency of FWI and extending its applicability to larger scales. This work presents the first 3D global synthetic dataset tailored for seismic wavefield modeling and full-waveform tomography, referred to as the Global Tomography (`GlobalTomo`) dataset. This dataset is uniquely comprehensive, incorporating explicit wave physics and robust geophysical parameterization at realistic global scales, generated through state-of-the-art forward simulations optimized for 3D global wavefield calculations. Through extensive analysis and the establishment of ML baselines, we illustrate that ML approaches are particularly suitable for global FWI, overcoming its limitations with rapid forward modeling and flexible inversion strategies. This work represents a cross-disciplinary effort to enhance our understanding of the earth's interior through physics-ML modeling.

## 1    INTRODUCTION

Global seismic tomography is a crucial yet intricate field within the earth sciences that facilitates understanding the earth's internal structure and dynamics. This discipline has wide-ranging applications, from the discovery of natural resources (Virieux and Operto, 2009) and the assessment of seismic hazards (Zang et al., 2021) to the exploration of our planet's evolutionary history (Dziewonski and Romanowicz, 2015). It utilizes seismic signals–earthquake-induced ground vibrations–recorded at surface stations to invert and map the earth's interior. The fundamental challenge in this field lies in the integration of forward modeling of seismic wave propagation with FWI, both of which are essential for accurately correlating seismic data with velocity structures (Romanowicz, 2003; Fichtner, 2010). However, accurately simulating seismic waves through the Earth's complex structures and inverting this data is challenging due to the heterogeneous nature of terrestrial media and the inherently ill-posed characteristics of FWI (Tromp, 2020; Lyu et al., 2021).

Modern numerical simulation methods, such as finite difference methods (Kelly et al., 1976), finite element methods (Zienkiewicz et al., 2005; Moczo et al., 2007), and spectral element methods (Komatitsch and Tromp, 2002a;b; Krischer et al., 2021), are fundamental in the forward modeling of seismic waves. Although these methods have proven effective, they impose significant computational demands, particularly when deploying higher-resolution models necessary for detailed geophysical interpretation. High-resolution imaging requires the simulation of high-frequency waves, which in turn necessitates smaller time steps and finer spatial discretization. This is essential to prevent numerical dispersion and maintain the stability of the simulations, as mandated by the Courant-Friedrichs-Lewy condition (De Moura and Kubrusly, 2013). Consequently, as the desired resolution increases, so does the computational burden, exponentially escalating the required computational resources and time, thereby introducing a substantial bottleneck (Li et al., 2022b).

The computational intensity of forward modeling significantly constrains the efficiency of FWI. FWI aims to iteratively refine a model of the earth's structure by minimizing the discrepancies

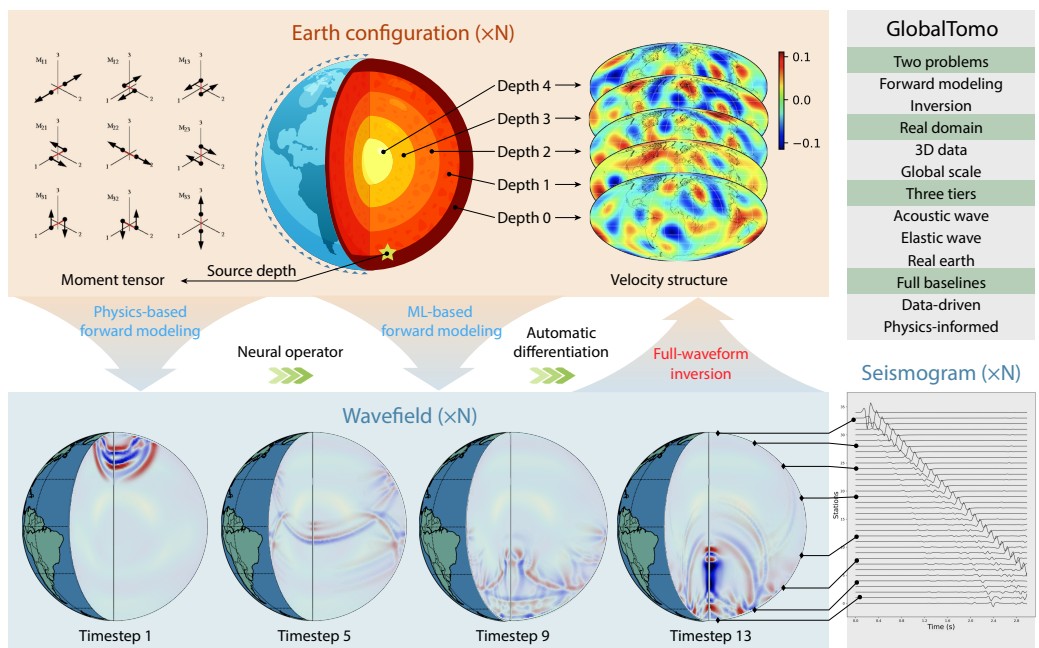

Figure 1: **Overview of `GlobalTomo`.** `GlobalTomo` is meticulously designed to tackle the pressing challenges associated with global seismic wavefield modeling and full-waveform inversion via cutting-edge physics-ML methodologies. In the forward modeling process, given specific source and velocity structures, the goal is to predict the wavefield at various time steps and the resulting seismograms at surface stations. The inversion process utilizes these seismograms as observational data to deduce the underlying velocity structures. Advanced ML techniques enhance these processes through neural operator learning and rapid automatic differentiation, substantially improving the efficiency of both forward modeling and inversion tasks.

between observed and simulated seismic data (Plessix, 2006). The prolonged computation times required for forward modeling restrict the exploration of the model parameter space and limit the number of feasible iterations (Bozdağ et al., 2016; Lei et al., 2020). This limitation is particularly problematic when attempting high-resolution inversions over extensive areas. Consequently, this situation highlights the critical need for more efficient methodologies in both seismic forward modeling (Nissen-Meyer et al., 2014) and inversion strategies (Adourian et al., 2023).

Recent advances in ML have shown significant promise in mitigating the aforementioned computational challenges in seismic tomography. Techniques such as Physics-Informed Neural Network (PINN) (Cuomo et al., 2022) and Neural Operator Learning (Lu et al., 2021; Li et al., 2020a; Wang et al., 2021) efficiently model complex physical systems by approximating solutions to the underlying Partial Differential Equations (PDEs). These methods have been successfully applied in fields including fluid dynamics prediction (Cai et al., 2021), new material discovery (Vasudevan et al., 2021), climate change modeling (O'Gorman and Dwyer, 2018; Pathak et al., 2022), and enhancing biomedical treatments (Liu et al., 2020). By significantly reducing computational time compared to traditional numerical simulations, these ML techniques enable more efficient high-resolution seismic modeling and inversion. Furthermore, modern ML approaches can generalize beyond trained scenarios, facilitating the exploration of previously uncharted Earth's interior.

However, the efficacy of ML methods in geophysical applications is heavily contingent upon the availability of extensive, high-quality training data. Currently, there is a significant lack of benchmark synthetic datasets specifically designed for global seismic tomography and the ML community. Although datasets to support FWI (Deng et al., 2022; Feng et al., 2023) have been introduced, they are mostly limited to subsurface exploration scenarios and do not fully represent the Earth's entire scale, which is crucial for high-resolution wave propagation modeling. This deficiency highlights a critical gap in the resources needed to advance ML applications in this area.

In response to these challenges, we introduce `GlobalTomo`, the first comprehensive 3D global full-waveform dataset specifically crafted for the ML community. `GlobalTomo` encompasses high-resolution seismic simulations that span from the earth's surface to its core. It features three distinct data tiers, integrating both acoustic and elastic wave equations to cater to a broad range of seismic phenomena. The dataset's scope varies from a local scale of a 1-km radius simulated at 20Hz to a global scale encompassing the entire earth's radius of 6,371 km at a 30-second period. We employ spherical harmonics to effectively parameterize velocity structures and account for variable seismic sources. Extensive benchmark analyses demonstrate `GlobalTomo`'s capabilities in fostering generalization, accelerating forward modeling, and addressing the inherent ill-posedness of seismic inversion tasks. By providing `GlobalTomo`, we aim to bridge the existing gap between geophysical sciences and the ML community, thereby enabling more precise and efficient seismic modeling and inversion processes through ML-driven innovations (Mousavi and Beroza, 2022).

## 1.1 Related Work

**Full-Waveform Inversion** FWI has been extensively applied across a variety of domains, including shallow crustal imaging (Tape et al., 2009; Górszczyk et al., 2017), continental tomography (Chen et al., 2007; Zhu et al., 2012), medical imaging (Schreiman et al., 1984; Wiskin et al., 2019), and nondestructive testing (Rao et al., 2016). In seismology, classical FWI employs numerical simulations to derive seismograms from an initial earth model and iteratively optimizes this model using the adjoint-state method (Tarantola, 1984; Tromp et al., 2005). While it has achieved significant success in both local and global inversions (Fichtner et al., 2009; French and Romanowicz, 2015), FWI still confronts substantial challenges such as high computational demands and susceptibility to local minima (Tromp, 2020). Emerging ML approaches, bolstered by comprehensive global seismic datasets, present promising avenues to address these challenges, potentially leading to more efficient and precise FWI solutions.

**Neural Operator Learning** Neural operator learning has garnered significant attention for solving parametric PDEs, as evidenced by a range of pioneering studies (Kovachki et al., 2023; Li et al., 2020b;a; Wang et al., 2021; Lu et al., 2021). Among these, DeepONet stands out for its flexibility in modeling mesh-free spaces and producing continuous solutions (Lu et al., 2021; Wang et al., 2021). Conversely, the Fourier Operator Learning (FNO) operates with mesh-based inputs and outputs yet efficiently models the frequency domain and demonstrates rapid convergence during training (Li et al., 2020a; Lu et al., 2022). Once trained, neural operators can swiftly predict solutions for new parameter sets with just a single forward pass. This capability is particularly advantageous in seismic applications, enabling the instant modeling of wave propagation under various velocity structures during inference. Such efficiency significantly streamlines the inversion process when compared to PINN methods (Song and Alkhalifah, 2021; Rasht-Behesht et al., 2022), offering a more direct and rapid approach to seismic data interpretation.

## 2 Dataset Construction

### 2.1 Problem Definition

The primary goal of forward modeling here is to solve the 3D acoustic and elastic wave equations governed by interior PDEs and boundary conditions detailed in appendix B.1. The objective is to predict the wavefield $\phi_{c,s}(p, t)$ at any spatial location $p = (x, y, z)$ and time $t$ across various velocity structures $c$ and source configurations $s$. This requires constructing and learning a nonlinear operator $G$, which maps velocity and source information into a wavefield output function, $G(c, s)$. The velocity structure $c$ is captured by sensor measurements at predetermined points, expressed as $c = [c(p_1), c(p_2), ..., c(p_m)]$. The source $s$ includes spatial locations and moment tensors. Given inputs $p$ and $t$, the function $G(c, s)$ computes the wavefield $\phi_{c,s}(p, t)$. This approach models wave propagation dynamics, aiding in accurate simulations and deeper geophysical understanding.

In particular, we monitor the displacements received by surface stations, known as seismograms, to infer the earth's underlying structures. The inversion problem can be formulated as:

$$c^* = \arg \min_c \|\mathbf{d} - \Phi(c, s)\|^2 + \lambda F(c, s). \tag{1}$$

Here, $\mathbf{d}$ represents the observed seismogram data. The function $\Phi(c, s)$ computes the seismograms based on the velocity structure $c$ and the source $s$. The term $F(c, s)$ is a regularization function that integrates prior knowledge and assumptions about the structure, while $\lambda$ is a coefficient that modulates the influence of the regularization term. The notation $\| \cdot \|^2$ denotes the L2 norm, used here to measure the misfit between the observed data and the predictions.

Traditional FWI relies on numerical simulations to compute $\Phi(c, s)$ and uses adjoint methods for optimizing $c$. Due to the intensive computational demand of these simulations, the optimization process is typically limited to a small number of iterations. We view ML methods as a potent tool to expedite the forward simulation process and enable more flexible optimization strategies. The solutions of modern ML approaches in addressing inversion problems are further discussed in appendix F.1.

## 2.2 MODEL CONFIGURATION

For generating global wavefield and seismogram data for our dataset, we employ AxiSEM3D (Leng et al., 2019), a forward simulator renowned for its efficiency in simulating global seismic wave propagation. AxiSEM3D is exceptionally adaptable, allowing customization to various model complexities (Leng et al., 2016). This simulator effectively utilizes the axisymmetry of global wavefields in a source-centered frame and extends calculations into the azimuthal Fourier domain. Further details are discussed in appendix B.2.

Our dataset comprises three tiers of increasing complexity, each supporting different training scales and applications. The first tier simulates acoustic wave propagation through a 1-km radius fluid sphere using a 20Hz mesh, providing a foundation in basic wave physics. The second tier extends these simulations to elastic wave propagation through an isotropic solid medium of the same scale, with source variations. This scale is comparable to 600 km continental simulations at a 30-second period, allowing exploration of complex P/S converted interactions. The third tier targets real-world earth applications, integrating acoustic and elastic simulations from the earth's surface to its core over a 30-second period, addressing planetary-scale wave propagation for detailed geophysical analyses.

Each model's 3D structure within these tiers is developed by introducing perturbations ranging from -10% to 10% to a 1D background model. These perturbations are meticulously designed to mirror the subtle yet significant 3D heterogeneity of the earth's interior, as uncovered by global tomographic studies (Dziewonski and Woodhouse, 1987; French and Romanowicz, 2014; Thrastarson et al., 2024). Such adjustments are crucial for demonstrating the impact of tomographic features on earth's geochemical and geodynamical processes, including mantle convection (van der Hilst et al., 1997), subduction of tectonic plates (Sun et al., 2014), and deep mantle heterogeneity such as the large low-shear-velocity provinces (Cottaar and Lekic, 2016). Further details on the configurations of each model are provided in appendix C.

**Parameterization** Perturbations within each defined layer are parameterized using real spherical harmonics (Dahlen and Tromp, 2020) up to degree 8, with radial values interpolated linearly between layers. Spherical harmonics, employed as the basis for parameterization, are mathematical functions that define a set of orthogonal functions on the sphere. These functions are particularly advantageous in geophysical applications because they can naturally represent the spherical geometry of the earth. Although some current mantle tomographic models offer higher resolution (Ritsema et al., 2011), spectrum analysis indicates that the significant power predominantly resides at lower degrees (Meschede and Romanowicz, 2015; Ritsema and Lekić, 2020). Consequently, our choice to limit structural parameterization to degree 8 in the published dataset effectively captures the predominant long-wavelength heterogeneity of the earth's interior (Su and Dziewonski, 1991). This approach reduces the number of parameters and enhances the solvability of this inverse problem.

## 2.3 DATA GENERATION

table 1 details simulation inputs and outputs. Using Latin Hypercube Sampling (LHS), we generated a set of random velocity structures and source configurations for extensive parameter coverage. Simulations were performed on a Dell PowerEdge R740 cluster with Intel Xeon Scalable processors (2,336 cores, up to 2.9GHz, 13,056GB RAM). Forward modeling for Acoustic and Elastic tiers

Table 1: **Simulation input and output size across three dataset tiers.** The upper section of the table provides basic information about the three data tiers, including specific parameters and configurations. The lower section of the table details the dimensions of time, spatial elements, and predicted seismic data for each tier, respectively.

| Category | Variables | Acoustic | | | Elastic | | | Real Earth | | |
|---|---|---|---|---|---|---|---|---|---|---|
| Input | Sample number | 10,000 | | | 30,000 | | | 10,000 | | |
| | Background model | Uniform | | | Uniform | | | PREM | | |
| | Time range | 0-3 s | | | 0-3 s | | | 0-6000 s | | |
| | Source | 0 | | | 6 | | | 9 | | |
| | Structure | 405 | | | 1215 | | | 5427 | | |
| **Dimension** | | **Time** | **Element** | **Seis** | **Time** | **Element** | **Seis** | **Time** | **Element** | **Seis** |
| Output | Wavefield | 15 | 58,368 | 1 | 15 | 58,368 | 3 | 20 | 263,520 | 3 |
| | Fourier wavefield | 15 | 3,648 | 16 | 15 | 3,648 | 48 | 20 | 16,470 | 48 |
| | Seismogram | 150 | 1,369 | 1 | 150 | 1,369 | 3 | 6,000 | 703 | 3 |
| | Storage | 88.54 GB | | | 657.28 GB | | | 1795.24 GB | | |

required about 2,800 and 8,400 CPU hours respectively, while seismic simulations for the Real Earth tier needed approximately 100,000 CPU hours. Despite the high computational demands, this method is more cost-effective than traditional adjoint inversions (Thrastarson et al., 2024).

Our dataset comprises two types of output data. The first type is surface seismogram data, which includes one-dimensional time series recordings of ground displacement at each surface station. These seismograms are observable globally and are crucial for the inversion of the earth's internal structure. The second type is seismic wavefield data, capturing the propagating seismic wavefield within the earth's interior. Although this spatially dense wavefield cannot be directly measured and used for traditional inversion, it reveals complex interactions between seismic waves and the earth's three-dimensional structure. The intricacies of the wavefield, which contain rich details, have recently gained attention in global wavefield studies (Leng et al., 2020) but are often too challenging to identify manually. We provide this intermediate wavefield data to facilitate the training of neural networks. For detailed configurations of each type of output data, please refer to appendix C. We compare the related dataset with `GlobalTomo` from various aspects in appendix D.

## 3 EXPERIMENTS

We experiment with several ML baselines as the general solution operator for the 3D acoustic and elastic wave equations to accompany `GlobalTomo`. These baselines, trained on the Acoustic and Elastic tiers, serve as practical workflows for forward modeling tasks, predicting wavefields and seismograms based on given velocity structures and source parameters. Wavefield predictions include 7 timesteps from 0 to 3 seconds, with each timestep consisting of 16 slices and 3,648 points per slice. Seismogram predictions cover a series of time series of length 150 for 1,369 stations. These forward modeling baselines are not only utilized to simulate data but also employed in inversion tasks to refine the velocity structures based on their predictions.

### 3.1 BASELINES

We implement our baselines using NVIDIA Modulus Sym (Hennigh et al., 2021), a framework designed for high-performance scientific computing.

**Mean Model (MM)** The MM baseline calculates the average results from the training data and uses these averages as predictions for the test data. Although this method does not capture complex data patterns, it serves as a useful benchmark for assessing the performance of more sophisticated models. Additionally, it provides insights into the overall distribution of the data.

**Multilayer Perceptron (MLP)** The MLP baseline consists of a 6-layer fully connected neural network, each layer having a hidden size of 500 and utilizing SiLU activation functions. This model takes velocity structures as inputs and predicts both wavefields and seismograms.

Table 2: **Quantitative results of baselines.** Performance of baseline models in forward modeling is evaluated on the test set using RL2 and R. Results include both mean and standard deviation values. The average inference time for each model is also presented.

| Wave | Model | Representation | Wavefield | | Seismogram | | Time |
|---|---|---|---|---|---|---|---|
| | | | RL2 ↓ | R ↑ | RL2 ↓ | R ↑ | (ms) |
| Acoustic | MM | Vector | 0.495±0.048 | 0.871±0.026 | 0.597±0.069 | 0.802±0.053 | - |
| | MLP | Vector | 0.356±0.035 | 0.937±0.012 | **0.446±0.051** | **0.896±0.026** | **1.771** |
| | H-Fourier | Vector | 0.397±0.054 | 0.917±0.066 | 0.594± 0.073 | 0.820±0.068 | 2.604 |
| | DeepONet | Point | 0.368±0.050 | 0.927±0.066 | 0.503±0.061 | 0.862±0.067 | 2.898 |
| | GNOT | Point | **0.300±0.014** | **0.954±0.004** | 0.564±0.041 | 0.829±0.026 | 2.985 |
| Elastic | MM | Vector | 1.000±0.001 | -0.006±0.113 | 1.000±0.001 | 0.001±0.116 | - |
| | MLP | Vector | **0.617±0.038** | **0.790±0.027** | **0.534±0.068** | **0.848±0.039** | **1.773** |
| | H-Fourier | Vector | 0.735± 0.061 | 0.689±0.048 | 0.613±0.067 | 0.790±0.049 | 2.397 |
| | DeepONet | Point | 0.682±0.035 | 0.734±0.031 | 0.565±0.065 | 0.826±0.043 | 2.775 |
| | GNOT | Point | 0.665±0.032 | 0.752±0.035 | 0.621±0.062 | 0.769±0.079 | 2.875 |

**Highway Fourier Net (H-Fourier)**   Considering the tendency of neural networks to favor low-frequency solutions (Rahaman et al., 2019), we explore the use of a H-Fourier. This model combines the gating mechanisms of Highway Networks (Srivastava et al., 2015) with the frequency domain learning capabilities of Fourier Neural Networks (Tancik et al., 2020). The H-Fourier is designed to handle deep learning tasks requiring both complex layer-to-layer transformations and frequency-specific learning, making it potentially more adept at managing intricate patterns in seismic data.

**DeepONet**   Instead of directly mapping velocity structures to fixed output grids, this approach involves building a framework that models both wave propagation and seismogram generation on a continuous spatial-temporal domain. We utilize the DeepONet (Lu et al., 2021), where the trunk net processes inputs of positions and times, and the branch net handles velocity structures. The outputs from both nets are combined through a dot product before passing through a linear output layer to produce the final prediction. To improve training efficiency and reduce memory, we developed an optimized method for parallel multiplication of the trunk and branch net outputs; see appendix E.2.

**Physics-Informed DeepONet (PIDO)**   While DeepONet is trained using traditional supervised learning, which might limit its generalization in high-resolution domains, we explore additional capabilities with PIDO (Wang et al., 2021). This model is designed to enforce physically consistent representations by incorporating interior and free-surface constraints during training. These constraints are integrated by calculating the necessary derivatives within the PDEs using automatic differentiation.

**General Neural Operator Transformer (GNOT)**   The GNOT (Hao et al., 2023) leverages the strengths of both neural operators and transformer architectures to effectively capture intricate spatial and temporal dependencies in data. By utilizing a self-attention mechanism, GNOT can efficiently model long-range interactions and handle high-dimensional inputs, enabling accurate predictions of dynamic systems like seismic wave propagation.

## 3.2   Results

### 3.2.1   Forward Modeling

**Generalization to unseen structures**   To investigate the ability of the baseline models to generalize across unseen seismic structures, we trained them using approximately 90% of the velocity structures available in our dataset. The remaining 10% of the structures were used as a test set to evaluate the models' performance. Quantitative assessments were carried out using Relative L2 Loss (RL2) and Person Correlation Coefficient (R) metrics.

Differences in baseline performance highlight the strengths of various modeling approaches. As detailed in table 2, MLP achieves high correlation coefficients and low RL2 values, demonstrating its efficacy in learning from vector-based representations. Observations from training indicate

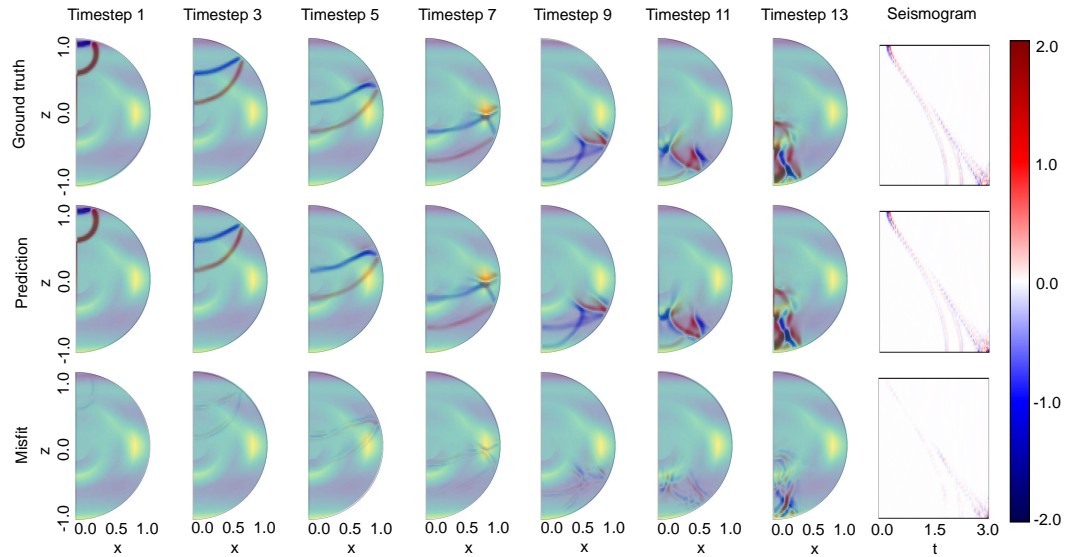

Figure 2: **Qualitative forward modeling prediction by MLP in the Acoustic tier**. A single slice is displayed. The source is located at $x = 0.00$ and $z = 0.80$. The background illustrates the velocity structure. The seismogram depicts the time series received by stations around the surface of this slice.

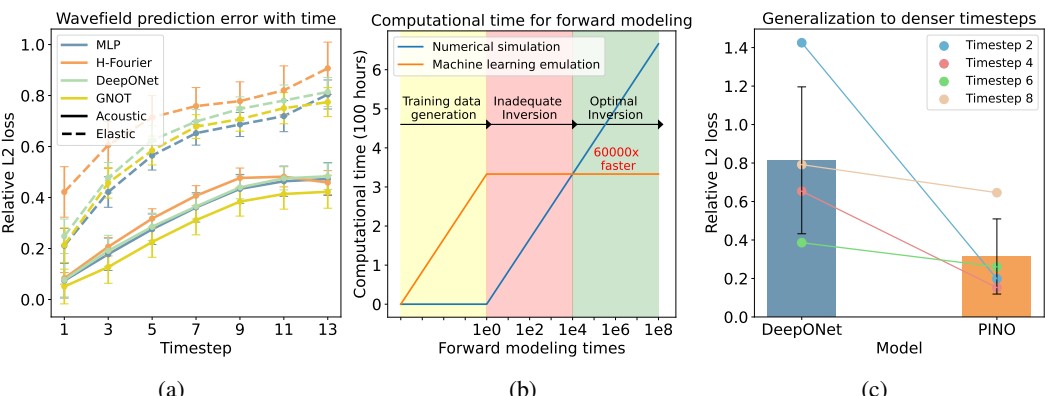

Figure 3: **Meta-analysis of forward modeling.** (a) RL2, used to quantify error in wavefield prediction, shows increasing trends over time across baseline models. (b) Once trained, ML emulation significantly outpaces numerical simulations in speed, facilitating more iterations in inversion processes. (c) DeepONet, when trained on timesteps 1, 3, 5, and 7, struggles to generalize to intermediate timesteps such as 2, 4, 6, and 8. Incorporating physical constraints during training improves model performance on these denser timesteps.

that the H-Fourier model reaches low training errors quickly, suggesting its proficiency in capturing frequency-based signals; however, it tends to overfit, resulting in limited generalization capabilities compared to the MLP. While vector-based methods excel with structured data, they face challenges in predicting arbitrary coordinates and time steps. In contrast, the point-based DeepONet and GNOT offer greater flexibility in generalizing across different coordinates and time steps, supporting mesh-free predictions. Nonetheless, it requires more computational resources during training to achieve comparable performance to the MLP. Further training details are available in appendix E. Qualitative evaluations, illustrated in fig. 2, corroborate these findings. The ML baselines closely match the actual data across several time steps, capturing the complex dynamics of wave propagation on unseen structures. As time advances, emulation errors tend to increase, reflecting the growing complexity of wave dynamics over extended periods, as depicted in fig. 3a.

**Complexity analysis**   The complexity of the acoustic and elastic waves comes from the intrinsic mechanical properties and mathematical formulation. For instance, the acoustic wavefield only has compressional (P) waves, while the elastic wavefield contains complex P/S converted energy. The performance of MM can serve as an indicator to reflect the dataset's complexity and distribution characteristics. Smaller error margins suggest more consistent patterns across velocity structures. The Elastic tier, evidenced by the poorer performance of MM, exhibits greater complexity than the Acoustic tier due to the intrinsic properties of the elastic wave equation and variations in source dynamics. The trend towards increased complexity is also corroborated by the performance metrics observed across different ML baselines.

**Forward modeling speed**   A single round of forward simulation using a numerical solver requires 120 seconds with 24 CPU cores. In contrast, ML approaches achieve forward modeling speeds typically ranging from 1 to 3 milliseconds on a single GPU, making them about 60,000 times faster than traditional methods. Note that the acceleration rate may vary with the degree of parallelism. While most computational costs for ML seismic modeling are incurred during the generation of training data, once trained, the neural operator can predict all wavefields in a single inference step. This capability underscores the potential of ML approaches for real-time seismic modeling and extensive forward modeling applications. For a comparative visualization, see fig. 3b.

**Generalization to higher temporal resolution**   To evaluate the flexibility of FWI in real-world scenarios through continuous wavefield prediction across the entire temporal domain, we tested the generalizability of a trained neural operator to higher resolution snapshots. Specifically, we utilized DeepONet, which accommodates mesh-free inputs. DeepONet was trained on a uniform velocity structure using acoustic waves at timesteps 1, 3, 5, and 7, and subsequently evaluated on a denser sequence from timesteps 1 to 8, within the same temporal range. The findings, illustrated in fig. 3c, show that DeepONet, being purely data-driven, initially struggled with adapting to this higher-resolution domain. To enhance its generalization capabilities, we incorporated physical constraints (detailed in appendix B.1) into its training regimen. This integration of physical principles significantly improved DeepONet's predictions on previously unseen timesteps, reducing its dependency on extensive labeled training data sets for developing a super-resolution model.

### 3.2.2   INVERSION

In this section, we explore how the generalization and acceleration capabilities of ML methods in forward modeling can significantly improve the inversion process. We present three strategies to address the challenges inherent in traditional FWI and enhance the efficiency of inversion operations.

**Gradient-based optimization**   Following traditional FWI methods, we initially set fixed parameters for established baselines and iteratively refine the velocity model to minimize the L1 misfit between the predicted and actual seismograms. Our optimization strategy employs 200 iterations of gradient descent using the LBFGS algorithm, with a learning rate of 0.08 and a history size of 30. A regularization term is integrated to promote stable convergence. We explore the effectiveness of gradient information from ML forward models in aiding the inversion optimization process. For each test structure, five initial points are randomly selected, and optimization is performed using the forward models pre-trained on the acoustic tier. The inversion performance is assessed using R across various structural scales with 80, 245, and 405 free parameters corresponding to degrees 4, 6, and 8, respectively. We show the optimization process of the best-performing model in fig. 4. The correlation between the inverted structure and the ground truth strengthens with each iteration step, confirming that ML-derived gradients can effectively guide the inversion process within 200 steps.

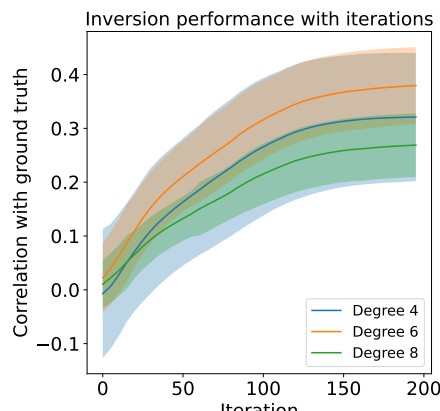

Figure 4: **The inversion optimization process.** Performance improves through optimization across 200 iterations. Higher degrees capture increasingly shorter-wavelength structures, enhancing model fidelity.

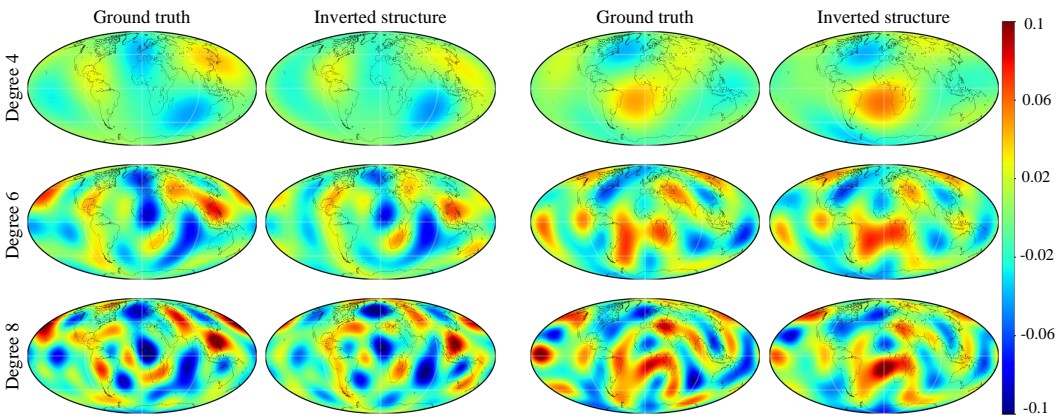

Figure 6: **Qualitative inversion results.** The results compare the inverted structure of the earth's interior, derived from acoustic data, with the ground truth across different spherical harmonic degrees. The color scale, transitioning from blue to red, indicates negative to positive perturbations respectively. For additional visualizations, please refer to appendix H.

**Sampling with multiple starting points** The initial conditions significantly influence the performance of inversion. To enhance inversion, we experimented with starting the optimization from multiple initial points. Specifically, we used ML forward modeling to generate 1,000 random starting points for each structure in the test set using LHS. Each starting point underwent optimization with the chosen algorithm, and the best result was selected for analysis. As illustrated in fig. 5, performance consistently improves with an increase in the number of starting points across various structural scales. This suggests that using more starting models could further boost inversion efficacy.

In summary, both increasing the number of optimization steps and using more starting models significantly enhance inversion performance. This underscores the ability of ML emulation to effectively tackle the challenges of ill-posedness and local minima—issues that are often problematic in traditional numerical solvers due to the high computational demands of extensive forward modeling. The use of ML approaches allows for more comprehensive exploration of solution spaces, which is critical for achieving accurate and robust inversion results.

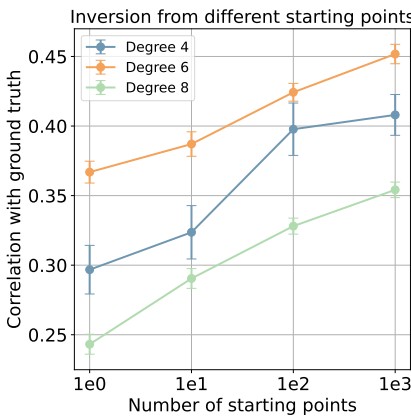

Figure 5: **Inversion with increasing starting points.** Correlation between the inverted models and the ground truth strengthens as the number of initial starting points increases, indicating improved robustness and accuracy of the inversion process.

**Direct inversion mapping** An alternative inversion strategy is to directly map observed seismograms to velocity structures. This method bypasses the conventional requirement for a forward ML model, potentially addressing the optimization challenges inherent in traditional FWI. We evaluated this approach by training the InversionNet-3D (Deng et al., 2022) with a 3D CNN backbone and an InversionMLP with 6 fully connected layers and 1000 hidden units, designed to input acoustic wave seismogram data and output velocity structures. Post-training, these models predicted reasonable velocity structures across various degrees in an unseen test set. The InversionMLP achieved the best average R of 0.826, MAE of 0.253, and RMSE of 0.335. Visualization of these results is provided in fig. 6. This direct inversion mapping approach offers promising insights into solving ML-based inversion problems more efficiently.

The quantitative inversion results for both optimization-based inversion and direct inversion can be found in table 3.

Table 3: **Quantitative inversion Results.** The optimization-based inversion utilizes pre-trained forward models. The direct inverse strategy predicts the inverted velocity structures directly from the observed seismograms.

| Model | Strategy | R ↑ | MAE ↓ | RMSE ↓ |
|---|---|---|---|---|
| MLP+L-BFGS | Optimization | 0.415 | 0.437 | 0.528 |
| DeepONet+L-BFGS | Optimization | 0.225 | 0.482 | 0.569 |
| H-Fourier+L-BFGS | Optimization | 0.161 | 0.618 | 0.750 |
| InversionNet-3D | Direct inverse | 0.794 | 0.272 | 0.356 |
| InversionMLP | Direct inverse | **0.826** | **0.253** | **0.335** |

### 3.2.3 DISCUSSION AND FUTURE WORK

- **Uniqueness of `GlobalTomo`.** The universality and representativeness of the dataset are critical for ensuring robust performance when applying the trained network to real-world data. In the realm of earth science, we benefit from concentrating on a single, unique object–the Earth–from its surface to its core. Unlike datasets from localized explorational or geological sites, which can vary significantly, our dataset is crafted to encapsulate the global attributes of the earth's interior. This focus enhances the dataset's applicability as all simulations and models are framed within a consistent setup. Future work could expand the scope by straightforwardly incorporating more degrees of spherical harmonics to explore the Earth's interior with greater detail.

- **Broader interest of `GlobalTomo`.** Beyond application in earth, our dataset is also crucial for advancing planetary sciences, especially for celestial bodies like Mars and the Moon. Seismic data for these planets are rare and costly to obtain. A pre-trained model that simulates seismic activity on these bodies can significantly improve our understanding of their interior.

- **Potential of ML.** ML presents a swift and powerful approach to addressing long-standing scientific challenges. Utilizing a finite set of simulated data that incorporates physics-inspired efficient representations, ML methods can perform global forward modeling and inversion across various structures and scales with minimal time investment. The accuracy and efficiency of these methods could be further enhanced by increasing the dataset size and adopting more advanced models. Furthermore, ML can be developed to reconcile differences between synthetic simulation outcomes and real-world data.

## 4 CONCLUSION

In this work, we introduce `GlobalTomo`, a comprehensive 3D global full waveform dataset for physics-ML research in forward modeling and FWI. By synthesizing high-fidelity seismic data across a range of scales and complexities with efficient spherical parameterization, `GlobalTomo` facilitates the advanced training and evaluation of ML models in the realm of seismic tomography. Our extensive evaluations confirm that ML-based approaches, supported by `GlobalTomo`, substantially enhance the efficiency and scalability of seismic wavefield modeling and FWI processes. Moreover, this dataset offers new possibilities for integrating machine learning with physics-based modeling. Through this cross-disciplinary endeavor, we anticipate that the convergence of advanced machine learning techniques and traditional geophysical methods will accelerate discoveries in earth science, promoting a deeper understanding of the Earth's interior dynamics and supporting critical applications such as resource discovery and hazard assessment.

**Societal Impacts** A deeper understanding of the earth's internal structure brings numerous societal benefits, including improved disaster preparedness and mitigation, safer urban planning and infrastructure development, more effective and sustainable management of natural resources, and environmental preservation. This research also fosters global scientific and technological collaboration, further amplifying its positive societal impacts.

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

CONTENTS OF SUPPLEMENTARY MATERIAL

## A    DATA AND CODE ACCESS

We will release our dataset and the documented code for data loading, training, and evaluation. We will continuously update and maintain both the code and data to ensure they reflect the latest advancements.

## B    PHYSICAL REPRESENTATIONS

### B.1    PHYSICAL CONSTRAINTS

We describe the physical constraints of the seismic wave propagation that can be used in the training process of neural networks. For the acoustic dataset tier, the wave propagation follows the acoustic wave equation in the fluid domain $\Omega_f$:

$$\frac{\partial_t^2 \chi}{\kappa} = \nabla \cdot (\frac{\nabla \chi}{\rho}), \quad in \quad \Omega_f. \tag{A1}$$

The output displacement $\mathbf{u} = \nabla \chi / \rho$, $\chi$ denotes the wave potential, $\rho$ represents the fluid density, and $\kappa$ represents the bulk modulus. The acoustic dataset has a pressure-free boundary condition that can be stated by:

$$\chi = 0, \quad on \quad \partial \Omega_f. \tag{A2}$$

The elastic wave propagation in the solid domain $\Omega_s$ follows:

$$\rho \partial_t^2 \mathbf{u} - \nabla \cdot (\mathbf{C} : \nabla \mathbf{u}) = 0, \quad in \quad \Omega_s. \tag{A3}$$

where $\mathbf{C}$ represents the elasticity tensor. The elastic dataset has a traction-free surface boundary condition:

$$\hat{\mathbf{n}} \cdot (\mathbf{C} : \nabla \mathbf{u}) = 0, \quad on \quad \partial \Omega_s. \tag{A4}$$

The Real Earth tier is a combination of these two scenarios. The seismic wave propagation in the fluid outer core follows eq. (A1) while in the resting solid places follows eq. (A3). This tier still has a traction-free boundary condition same as eq. (A4). The solid-fluid coupling is controlled by the displacement- and stress-continuity condition on the fluid-solid interface $\Sigma$:

$$\hat{\mathbf{n}} \cdot \mathbf{u} = \chi_r / \rho \quad , \quad \hat{\mathbf{n}} \cdot (\mathbf{C} : \nabla \mathbf{u}) = \partial_t^2 \chi \hat{\mathbf{n}} \quad on \quad \Sigma. \tag{A5}$$

### B.2    EXPANSION TO AZIMUTHAL FOURIER DOMAIN

Describing the 3D earth is memory-consuming. Inspired by the smoothness and symmetry found in the physical properties of the earth, representing the wavefield in the Fourier domain can achieve a more efficient description. If the time-dependent equations of motion can be generalized in computational domain $\Omega$ as:

$$\mathscr{L} \mathbf{u} = \mathbf{f} \tag{A6}$$

The displacement $\mathbf{u}$, operator $\mathscr{L}$, and external forces $\mathbf{f}$ will be characterized to the required azimuth resolution defined by $n_u$:

$$\mathbf{u}(r, \theta, \phi; t) = \sum_{|\alpha| \le n_u} \mathbf{u}^\alpha(r, \theta; t) \exp(i\alpha\phi) \tag{A7}$$

$$\mathscr{L}(r, \theta, \phi; t) = \sum_{|\beta| \le n_u} \mathscr{L}^\beta(r, \theta; t) \exp(i\beta\phi) \tag{A8}$$

$$\mathbf{f}(r, \theta, \phi; t) = \sum_{|\gamma| \le n_u} \mathbf{f}^\gamma(r, \theta; t) \exp(i\gamma\phi) \tag{A9}$$

Thus this method reduces the dimension of azimuth that is needed to solve a true 3D problem and saves huge computation costs while also ensuring the accuracy of the solution. If we apply the structural parameterization of spherical harmonics and determine the highest degree of the model, AxiSEM3D currently has the best theoretical efficiency for calculating their global wavefields, which only need $2n_u$ times the computation of a 2D problem. In `GlobalTomo`, we provide the Fourier series of the wavefield for building more robust forward models with efficient representation in the future.

## B.3 Weak Formulation and Dimension-Reduced Form

The weak formulation of the elastic equation can be obtained by taking dot product with an arbitrary test function $\mathbf{w}$ and integrating by parts over the volume $\Omega_s$:

$$\int_{\Omega_s} \left( \rho \partial_t^2 \mathbf{u} \cdot \mathbf{w} + \nabla \mathbf{u} : \mathbf{C} : \nabla \mathbf{w} \right) d\mathbf{x}^3 = \int_{\Omega_s} \mathbf{f} \cdot \mathbf{w} d\mathbf{x}^3 \tag{A10}$$

The stress-free boundary condition is thus naturally merged into the weak formulation of the constraint.

Its corresponding dimension-reduced form in a 2D domain $D$ can be written down as:

$$\sum_{|\alpha| \leq n_u} [\langle \rho^{-(\alpha+\beta)} \partial_t^2 \mathbf{u}^\alpha, \mathbf{w}^\beta \rangle_D + a_D^{-(\alpha+\beta)}(\mathbf{u}^\alpha, \mathbf{w}^\beta)] = \langle \mathbf{f}^{-\beta}, \mathbf{w}^\beta \rangle_D \tag{A11}$$

$$\forall \mathbf{w}^\beta \ in \ D, \quad \forall \beta \in \{-n_u, ..., n_u\}$$

$$\langle \mathbf{u}^\alpha, \mathbf{w}^\beta \rangle_D := \int_D u_i^\alpha w_i^\beta s ds dz \tag{A12}$$

$$a_D^\gamma(\mathbf{u}^\alpha, \mathbf{w}^\beta) := \int_D u_{i,j}^\alpha C_{ijkl}^\gamma w_{k,l}^\beta s ds dz \tag{A13}$$

where the azimuthal expansion of $\mathbf{u}$, $\mathbf{f}$, $\rho$, and $C$ follows:

$$\mathbf{u} = \sum_{|\alpha| \leq n_u} \mathbf{u}^\alpha(s, z; t) \Psi^\alpha(\phi) \tag{A14}$$

$$\mathbf{f} = \sum_{|\alpha| \leq n_u} \mathbf{f}^\alpha(s, z; t) \Psi^\alpha(\phi) \tag{A15}$$

$$\rho = \sum_{|\alpha| \leq n_u} \rho^\alpha(s, z; t) \Psi^\alpha(\phi) \tag{A16}$$

$$C_{ijkl} = \sum_{|\alpha| \leq n_u} C_{ijkl}^\alpha(s, z; t) \Psi^\alpha(\phi) \tag{A17}$$

Here, $\Psi^\alpha(\phi)$, $\alpha \in \{-n_u, ..., n_u\}$ denote the azimuthal Fourier modes,

$$\Psi^\alpha(\phi) := \exp(\mathrm{i}\alpha\phi), \mathrm{i} = \sqrt{-1}. \tag{A18}$$

# C Earth Configuration Details

## C.1 Concept Illustration

- **Source:** Described by location and moment tensor, a source of seismic energy can be either natural or artificial.

- **Velocity Structure:** Refers to the distribution of seismic wave velocities within the earth's interior, determined by the physical and chemical properties of the materials.

- **Wavefield:** The spatial pattern of wave propagation within the earth.

- **Seismogram:** The recordings of ground motions received by stations on the surface.

- **Forward Modeling:** The process of computing numerical solutions of the wave propagation from a defined model of the earth's interior.

- **Full Waveform Inversion (FWI):** An advanced seismic inversion technique that leverages the entire seismic waveform signals to develop high-resolution models of the earth's interior.

## C.2 THREE DATASET TIERS

We introduce the details of the three tiers generated by AxiSEM3D. The simulator is licensed under the MIT license at https://github.com/AxiSEMunity/AxiSEM3D.

Our dataset includes three tiers of synthetic seismic data. The first tier models acoustic wave propagation through a fluid sphere with a 1-km radius using a 20 Hz mesh. The second tier models elastic wave propagation through an isotropic solid medium of the same scale, with variations in the source. The third tier is designed for real-world earth applications, integrating both acoustic and elastic simulations over a 30-second period.

We export the 3D velocity perturbations in Cartesian coordinates to facilitate rapid interpolation and retain the original spherical harmonics coefficients for use in network training. To ensure the adherence to physical constraints, we disable attenuation.

**Acoustic Ball**    The acoustic ball model is designed on a 1000 m radius ball with pure fluid medium and a mesh accurate to 20 Hz. This set of models has a pressure-free boundary condition. The wave propagation in the model obeys the acoustic wave equation. The medium only has the P-wave velocity (vp) attribute. The 1D background model has a uniform vp value of 1.0 km/s across the ball. The 3D superimposed velocity perturbation is parameterized by spherical harmonics up to degree 8, leading to 81 spherical parameters per layer, multiplied by 5 layers, resulting in 405 free parameters in total.

The source is located at a fixed 200 m depth at the north pole with a fixed fluid pressure source.

**Elastic Ball**    The elastic ball model is also designed on a 1000 m radius ball with pure elastic medium and a mesh accurate to 20 Hz. This set of models has a stress-free boundary condition. The wave propagation in the model obeys the elastic wave equation. The isotropic medium only has three independent attributes: P-wave velocity (vp), S-wave velocity (vs), and density (rho). The 1D background model has a uniform vp value of 1.5 km/s, vs value of 1.0 km/s, and rho of 1.0g/cm$^3$ across the ball. The 3D superimposed velocity perturbation is parameterized by spherical harmonics up to degree 8, leading to 81 spherical parameters per layer, multiplied by 5 layers and 3 medium attributes, resulting in 1215 free parameters in total.

The source is located at a fixed 200 m depth at the north pole with a variable moment tensor source mechanism. This introduces 6 extra free parameters in source terms.

**Real Earth**    This set of models is designed for the ultimate real-earth applications. We utilize the isotropic PREM (Dziewonski and Anderson, 1981) background model and a global mesh with a period accurate to 30 seconds that encompasses various earth's layers including the crust, mantle, outer core, and inner core, each characterized by their unique seismic attributes. The Real Earth tier has a stress-free boundary condition. The 3D velocity perturbations are imposed on the earth's internal 23 discontinuities, resulting in a total of 5427 free parameters up to degree 8 across the whole earth.

The source is located at stochastic depths between 0 - 800 km at arbitrary geographical coordinates with a variable moment tensor source mechanism. This introduces 9 extra free parameters in source terms.

**Surface Seismogram Output**    The seismogram data contains a one-dimensional time series recording ground displacement at each surface station. Each trace is low-pass filtered to 30 seconds to avoid numerical noise. We place such stations evenly along latitude and longitude to represent a spatially abundant coverage. The actual station deployments limit the real seismic data coverage, however, this issue can be considered by dumping part of the data during the application. The seismogram data can be observed globally and be used in the inversion of the earth's internal structure.

**Seismic Wavefield Output**    The seismic wavefields are saved in discrete 15 or 20 time snapshots, (total time = 3 s or 6,000 s), represented by 16 azimuthal slices with 3,648 or 16,470 (from surface to the core) element points in each. Besides, we also saved the azimuthal Fourier coefficients of the global wavefields for a continuous and compact representation in three dimensions. The wavefields and the seismograms are saved as HDF5 files for fast loading.

## D COMPARISON WITH OTHER RELATED DATASETS

We contrasts the GlobalTomo dataset with prior works across several key aspects in table A1. The comparison highlights the differences in scale, data utilization, parameterization, and simulation methods. GlobalTomo is first designed for global-scale inversion, incorporating the entire wavefield and receiver data, utilizing spherical harmonics for structural parameterization, and employing spectral element methods (SEM) for efficient simulation. In contrast, prior works (e.g., OpenFWI (Deng et al., 2022), E-FWI (Feng et al., 2024)) typically focus on local-scale applications, use only receiver data, rely on mesh representation, and often employ finite difference methods (FDM) for simulation. DiTing (Zhao et al., 2023) and SeisBench (Woollam et al., 2022) are particularly designed for phase picking, denoising, and earthquake detection instead of earth tomography.

Table A1: **Comparison with prior related datasets.**

| Category | Perspective | OpenFWI | E-FWI | DiTing | SeisBench | GlobalTomo |
|---|---|---|---|---|---|---|
| **Scale** | Global | | | | ✓ | ✓ |
| | Local | ✓ | ✓ | ✓ | ✓ | |
| **Wave type** | Acoustic | ✓ | | | | ✓ |
| | Elastic | | ✓ | ✓ | ✓ | ✓ |
| **Data utilization** | Wavefield | | | | | ✓ |
| | Receiver | ✓ | ✓ | ✓ | ✓ | ✓ |
| **Structural parameterization** | Sph. harmonics | | | | | ✓ |
| | Mesh | ✓ | ✓ | | | |
| **Simulation method** | FDM | ✓ | ✓ | | | |
| | SEM | | | | | ✓ |

## E BASELINE MODEL DETAILS

### E.1 EVALUATION METRICS

The RL2 metric is particularly valuable in scenarios where it's crucial to evaluate the performance of predictive models across datasets of varying scales or inherent variability. By normalizing the Mean Squared Error (MSE) with the variance of the actual values, this metric effectively renders the error measurement scale-independent. The formulation is as follows:

$$\text{Relative } L_2 \text{ Loss} = \sqrt{\frac{\sum_{i=1}^{n}(y_i - \hat{y}_i)^2}{\sum_{i=1}^{n}(y_i - \overline{y})^2}} \tag{A19}$$

where $y_i$ and $\hat{y}_i$ denote the actual and predicted values of the $i$-th observation, respectively. $\overline{y}$ is the mean of all actual values in the dataset, calculated as $\frac{1}{n}\sum_{i=1}^{n} y_i$.

We also use the Person Correlation Coefficient (R) to measure the similarity between the prediction and ground truth. The R is given by:

$$R = \frac{\sum(y_i - \overline{y})(\hat{y}_i - \overline{\hat{y}})}{\sqrt{\sum(y_i - \overline{y})^2 \sum(\hat{y}_i - \overline{\hat{y}})^2}} \tag{A20}$$

### E.2 EFFICIENT DEEPONET DESIGN

The original DeepONet takes paired velocity structure and spatial-temporal points for the trunk network and branch network. The output of DeepONet is calculated by combining the outputs from the branch and trunk networks, typically through a multiplication operation. Specifically, if the output from the trunk network is $a = [a_1, a_2, \ldots, a_d]$ and the output from the branch network is $b = [b_1, b_2, \ldots, b_d]$, then the predicted value $\hat{\phi}(p, t)$ at output position $p$ and time $t$ is given by:

$$\hat{\phi}(p,t) = \sum_{i=1}^{d} a_i \cdot b_i(p,t) \tag{A21}$$

Here, $a_i$ and $b_i(p,t)$ are the features obtained from the branch and trunk networks, respectively, and their product followed by summation forms the estimate of the output function.

During training, this network is optimized to minimize the error between the true output $\phi(p,t)$ and the predicted output $\hat{\phi}(p.t)$, typically using the mean squared error as the loss function:

$$\text{Loss} = \frac{1}{n} \sum_{j=1}^{n} \left( \phi(p_j, t_j) - \hat{\phi}(p_j, t_j) \right)^2 \tag{A22}$$

where $n$ is the number of output points.

However, this is memory-consuming since the velocity structures are duplicated for each point. Indeed, one velocity structure can be paired with a group of spatial and temporal points $P, T$ simultaneously. Inspired by (Lu et al., 2022), we employ a more efficient DeepONet that can predict the whole 3D outputs just using one forward of branch net. The calculation of the output function now is given by:

$$\hat{\Phi}(P,T) = \sum_{i=1}^{d} a_i \cdot B_i(P,T) \tag{A23}$$

where $B_i(P,T)$ indicates a vector of the i-th latent feature from a batch of points in the same velocity structure. $\hat{\Phi}(P,T)$ indicates a group of predicted output in the same structure. Suppose the input dimensions of the velocity structure and the spatial-temporal point are $m$ and 4, respectively, the memory usage in the modified DeepONet can be $\frac{4+m}{4+m/n}$ times smaller than the original one. The training speed is about 3 times faster.

### E.3 HYPERPARAMETERS

For reproducibility, we present the training configuration of baselines in table A2 and table A3. We employ the SiLu activation function and use the Adam optimizer for parameter tuning. The initial learning rate is set at 0.0003, and it decreases exponentially by a factor of 0.95 every 1000 steps. Both the MLP and H-Fourier models consist of six layers with 500 neurons each. The DeepONet model features six layers of 600 neurons each, and includes trunk and branch projection layers, each with 1000 neurons. The final output is computed via a dot product of the latent trunk and branch vectors. The GNOT model has 4 attention layers with a hidden size of 256.

The MLP and H-Fourier models are trained on a single A800 SXM4 80GB GPU. In contrast, Deep-ONet and GNOT require 8 A800 SXM4 80GB GPUs for training due to the substantial increase in data size associated with point-based prediction.

Table A2: **Wavefield model details.**

| Wave type | Model | Layer num | Hidden size | Training steps | Input size | Output size |
|-----------|-----------|-----------|-------------|----------------|------------|-------------|
| Acoustic | MLP | 6 | 500 | 10000 | 405 | 408576 |
| | H-Fourier | 6 | 500 | 3000 | 405 | 408576 |
| | DeepONet | 6 | 600 | 100000 | 405+3 | 1 |
| | GNOT | 4 | 256 | 200000 | 405+3 | 1 |
| Elastic | MLP | 6 | 500 | 30000 | 1221 | 408576 |
| | H-Fourier | 6 | 500 | 7000 | 1221 | 408576 |
| | DeepONet | 6 | 600 | 40000 | 1221+3 | 1 |
| | GNOT | 4 | 256 | 300000 | 1221+3 | 1 |

Table A3: **Seismogram model details.**

| Wave type | Model | Layer num | Hidden size | Training steps | Input size | Output size |
|---|---|---|---|---|---|---|
| Acoustic | MLP | 6 | 500 | 10000 | 405 | 205350 |
| | H-Fourier | 6 | 500 | 3000 | 405 | 205350 |
| | DeepONet | 6 | 600 | 100000 | 405+3 | 1 |
| | GNOT | 4 | 256 | 200000 | 405+3 | 1 |
| Elastic | MLP | 6 | 500 | 10000 | 1221 | 205350 |
| | H-Fourier | 6 | 500 | 4000 | 1221 | 205350 |
| | DeepONet | 6 | 600 | 100000 | 1221+3 | 1 |
| | GNOT | 4 | 256 | 300000 | 1221+3 | 1 |

## F DISCUSSION

### F.1 COMPARISON OF MODERN ML-BASED INVERSION

Here we discuss the difference between modern ML-based inversion approaches. The two most dominant schools are PINN and neural operator learning.

We first recall the formulation of the inversion problem.

$$c^* = \arg\min_c \|\mathbf{d} - \Phi(c, s)\|^2 + \lambda F(c, s) \tag{A24}$$

In comparing the inversion strategies of PINN and operator learning, we observe distinct paths of optimization. The PINN-based approach seeks to optimize the model parameters $c$ in conjunction with the forward modeling function $\Phi$ (Rasht-Behesht et al., 2022). Thus, the $c$ and $\Phi$ are closely coupled and the inversion of different structures requires training from scratch. This method limits the model's ability to generalize across different velocity structures.

In contrast, the operator learning approach addresses this limitation by initially learning a universal operator to model forward problems across all $c$ (Yang et al., 2023). Subsequently, it fine-tunes the model parameters $c$ for each specific case using the pre-trained $\Phi$. Once trained, the neural operator can achieve inversion on any structures in inference time.

Our study adopts the latter approach to achieve fast and flexible inversion. We study both the sampling-based and optimization-based methods when using the pre-trained neural operator to solve the inversion problem.

### F.2 IS PREDICTING WAVEFIELD NECESSARY FOR INVERSION?

As shown in section 3.2.2, direct inverse mapping demonstrates better inversion performance than the optimization-based method. We want to discuss whether predicting wavefield is still necessary from the aspects of complexity, interpretability, and flexibility.

**Complexity** Direct ML inversion methods offer an efficient alternative to traditional optimization-based approaches by using neural networks to approximate the inverse mapping from data to subsurface properties directly. This method significantly cuts down on computational time and resources, simplifying the traditional iterative inversion process into a single step. In contrast, splitting the FWI process into two stages might increase complexity regarding the implementation and integration of different computational frameworks or software tools. Additionally, more stages could lead to greater cumulative errors.

**Interpretability** Predicting the wavefield explicitly as part of the inversion process can enhance the interpretability of the results. In FWI, the aim is to reconstruct the subsurface properties using wave propagation data. Although direct inverse mapping can produce a result of the inferred structure, it doesn't explicitly model the intermediate process and tends to lack transparency in its operations. By decomposing the inversion into two distinct phases—predicting the forward wavefield and then optimizing the subsurface model—researchers can trace the effects of specific inputs

or changes in model parameters on the outcomes. This division aligns closely with traditional FWI methods where each step of the process can be examined and understood separately. Furthermore, explicitly modeling the wavefield allows for a more detailed examination of the physical accuracy and stability of the inversion process, enabling researchers to validate and refine the underlying physical models being used.

**Flexibility** Predicting the wavefield explicitly provides opportunities to apply a broader range of optimization strategies tailored to specific challenges or characteristics of the data. For example, if the predicted wavefield is treated as a separable component, researchers can implement advanced gradient methods such as adaptive step-sizing or preconditioning. These techniques can address specific dynamics of wave propagation in different geological settings, potentially enhancing the accuracy and efficiency of the inversion. Preconditioning, in particular, can help tackle issues like the ill-posed nature of the inversion problem or the presence of sharp local minima by altering the optimization landscape.

## F.3 Connection with Other Fields

Similar to the challenges of interpreting the human mind in the field of intuitive physics, the task of inverting Earth's structure involves comparable approaches and discussions.

Both seismic inversion and intuitive physics deal with inverse problems—inferring causes from effects. Seismic inversion back-calculates to infer the Earth's interior from surface measurements (Tromp, 2020). Similarly, intuitive physics tries to deduce the cognitive rules or mental models that people use to predict physical outcomes from their observed behavior in experimental settings (Kubricht et al., 2017).

Regarding methodologies, both fields heavily utilize computational models. In seismic studies, numerical methods and statistical models predict how geological features influence seismic waveforms (Tarantola, 1984; Tromp et al., 2005). Similarly, in intuitive physics, physics engines simulate how humans might predict physical interactions in their surroundings (Battaglia et al., 2013; Li et al., 2023). Machine learning, particularly neural networks, plays a significant role in both areas. In seismic inversion, neural networks can be trained to recognize patterns in seismic data that correlate with specific subsurface structures (Yang et al., 2023). In intuitive physics, neural networks model human prediction and reasoning processes about physical laws, learning to mimic human judgment (Li et al., 2022a).

Uncertainty management is crucial in both domains. In seismic inversion, uncertainties arise from limited sensor coverage and noise in data. In intuitive physics, uncertainties stem from the variability in human perception and cognitive biases. Both fields develop methodologies to handle these uncertainties, often through probabilistic models (Tromp, 2020; Battaglia et al., 2013).

# G Datasheet

## G.1 Motivation

- **For what purpose was the dataset created?** `GlobalTomo` was created to promote the physics-ML research on solving seismic full waveform modeling and earth structure inversion.

- **Who created the dataset (e.g., which team, research group) and on behalf of which entity (e.g., company, institution, organization)?** The dataset was created by a group of geophysical scientists and ML researchers.

## G.2 Composition

- **What do the instances that comprise the dataset represent (e.g., documents, photos, people, countries)?** `GlobalTomo` contains many simulated wavefield sequences and seismogram time series documented as arrays in HDF5 files.

- **How many instances are there in total (of each type, if appropriate)?** The acoustic tier has 10,000 cases of simulation results. The elastic tier has 30,000 cases and the real elastic tier has 10,000 cases.

- **Does the dataset contain all possible instances or is it a sample (not necessarily random) of instances from a larger set?** Yes, the dataset contains all possible instances for training and evaluation.

- **What data does each instance consist of?** Each instance consists of inputs and outputs. The inputs have velocity structures and source parameters. The outputs contains three types of features including the wavefield, the Fourier wavefield, and the seismogram. The wavefield includes a sequence of snapshots and each snapshot consists of 16 slices, together with the 3D coordinates and timesteps. The Fourier wavefield resembles the wavefield but presents each 3D snapshot in terms of 16 Fourier series. The seismogram consists of time series data from multiple stations, which also provide the 3D coordinates of each station and corresponding timesteps.

- **Is there a label or target associated with each instance?** Yes, each instance has clearly defined input and output variables.

- **Is any information missing from individual instances?** No.

- **Are there recommended data splits (e.g., training, development/validation, testing)?** We recommend using 90% of the data for training/validation and the remaining 10% for testing.

- **Are there any errors, sources of noise, or redundancies in the dataset?** The velocity structures are replicated across various features to facilitate data loading and training. Given the small size of these structures, the redundancy in storage can be considered negligible.

- **Is the dataset self-contained, or does it link to or otherwise rely on external resources (e.g., websites, tweets, other datasets)?** The dataset is self-contained.

- **Does the dataset contain data that might be considered confidential** No.

- **Does the dataset contain data that, if viewed directly, might be offensive, insulting, threatening, or might otherwise cause anxiety?** No.

### G.3 COLLECTION PROCESS

- **How was the data associated with each instance acquired?** Was the data directly observable (e.g., raw text, movie ratings), reported by subjects (e.g., survey responses), or indirectly inferred/derived from other data (e.g., part-of-speech tags, model-based guesses for age or language)? The data was collected from a set of simulations using a 3D global seismic model named AxiSEM3D.

- **What mechanisms or procedures were used to collect the data (e.g., hardware apparatuses** or sensors, manual human curation, software programs, software APIs)? We utilized a cluster of CPUs to run the AxiSEM3D.

- **Who was involved in the data collection process (e.g., students, crowdworkers, contractors) and how were they compensated (e.g., how much were crowdworkers paid)?** Geophysics researchers in the author list were involved in the data collection process and no crowdworkers were involved.

### G.4 PREPROCESSING/CLEANING/LABELING

- **Was any preprocessing/cleaning/labeling of the data done (e.g., discretization or bucketing, tokenization, part-of-speech tagging, SIFT feature extraction, removal of instances, processing of missing values)?** We examine the failure cases in simulation and remove them from training and evaluation. The data is processed into HDF5 files with labeled variables for easy loading.

- **Was the "raw" data saved in addition to the preprocessed/cleaned/labeled data (e.g., to support unanticipated future uses)?** Yes, the raw data is saved in our cluster.

- **Is the software that was used to preprocess/clean/label the data available?** Yes, we uploaded the code on GitHub.

## G.5 USES

- **Has the dataset been used for any tasks already?** No.

- **What (other) tasks could the dataset be used for?** The dataset could be used for seismic forward modeling, earth structure inversion, and source inversion.

- **Is there anything about the composition of the dataset or the way it was collected and preprocessed/cleaned/labeled that might impact future uses?** No.

- **Are there tasks for which the dataset should not be used?** No.

## G.6 DISTRIBUTION

- **Will the dataset be distributed to third parties outside of the entity (e.g., company, institution, organization) on behalf of which the dataset was created?** The dataset is open to the public.

- **How will the dataset be distributed (e.g., tarball on website, API, GitHub)?** The dataset will be distributed on Google Drive and the code will be published on GitHub.

- **Will the dataset be distributed under a copyright or other intellectual property (IP) license, and/or under applicable terms of use (ToU)?** The dataset will be distributed under the CC BY-NC-SA license.

- **Have any third parties imposed IP-based or other restrictions on the data associated with the instances?** No.

- **Do any export controls or other regulatory restrictions apply to the dataset or to individual instances?** No.

## G.7 MAINTENANCE

- **Who will be supporting/hosting/maintaining the dataset?** The `GlobalTomo` group will support, host, and maintain the dataset.

- **Is there an erratum?** No.

- **Will the dataset be updated (e.g., to correct labeling errors, add new instances, delete instances)?** Yes, the dataset will be continuously updated if there is a necessity to improve accuracy and other related information. The updates will be released on the website.

- **If the dataset relates to people, are there applicable limits on the retention of the data associated with the instances (e.g., were the individuals in question told that their data would be retained for a fixed period of time and then deleted)?** The dataset is not related to people.

- **Will older versions of the dataset continue to be supported/hosted/maintained?** Yes, older versions of the dataset will continue to be supported, maintained, and hosted.

- **If others want to extend/augment/build on/contribute to the dataset, is there a mechanism for them to do so?** Yes, the contributor can contact us through email.

# H MORE VISUALIZATION

## H.1 FORWARD MODELING

We further present the visualization of forward modeling on four novel velocity structures to demonstrate that the ML baselines are capable of discerning differences in wave propagation. We display snapshots at timesteps 5, 7, 9, 11, and 13, illustrating that the variance in wavefields grows over time. Refer to figs. A1 to A5 for detailed visualizations.

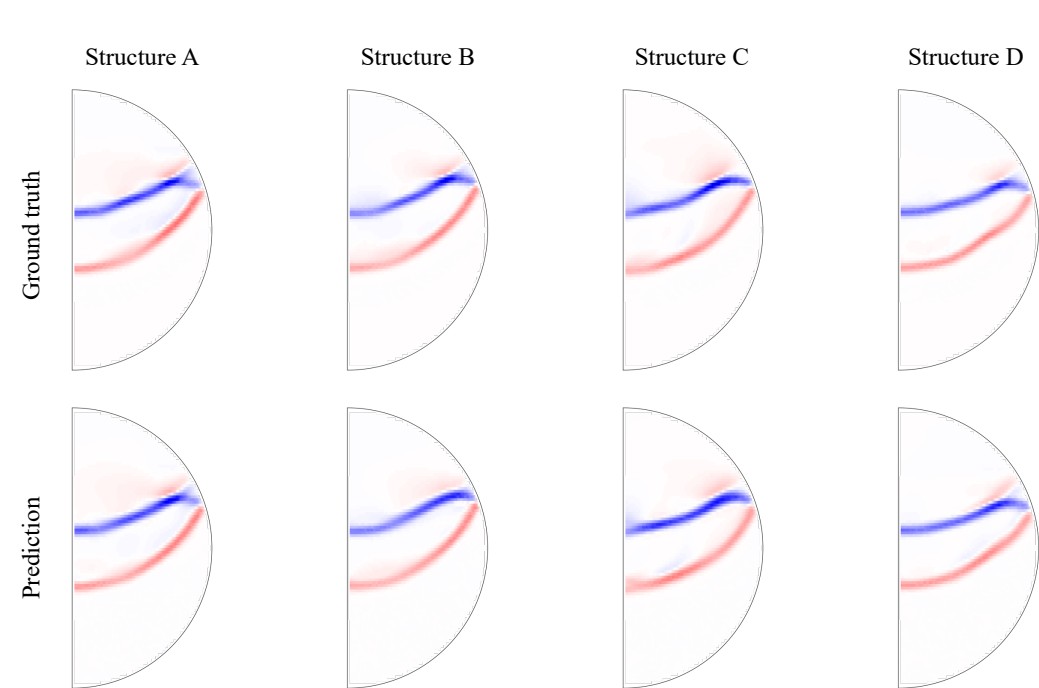

Figure A1: **Snapshot 5 of forward modeling.** We show the ground truth and predicted wavefield snapshot within four velocity structures: A, B, C, and D.

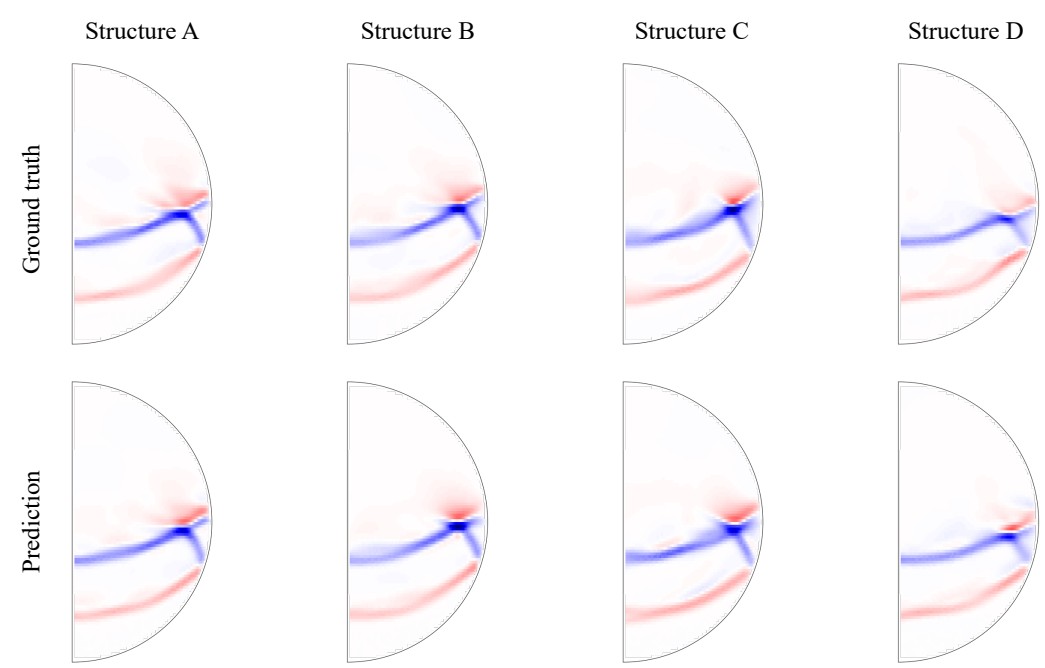

Figure A2: **Snapshot 7 of forward modeling.** We show the ground truth and predicted wavefield snapshot within four velocity structures: A, B, C, and D.

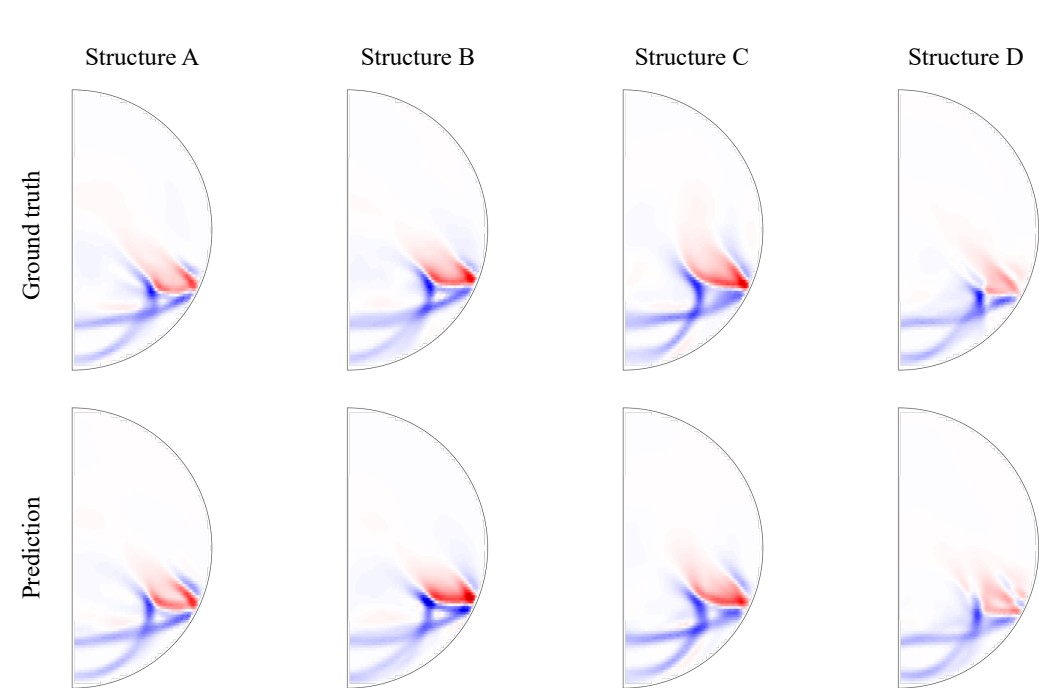

Figure A3: **Snapshot 9 of forward modeling.** We show the ground truth and predicted wavefield snapshot within four velocity structures: A, B, C, and D.

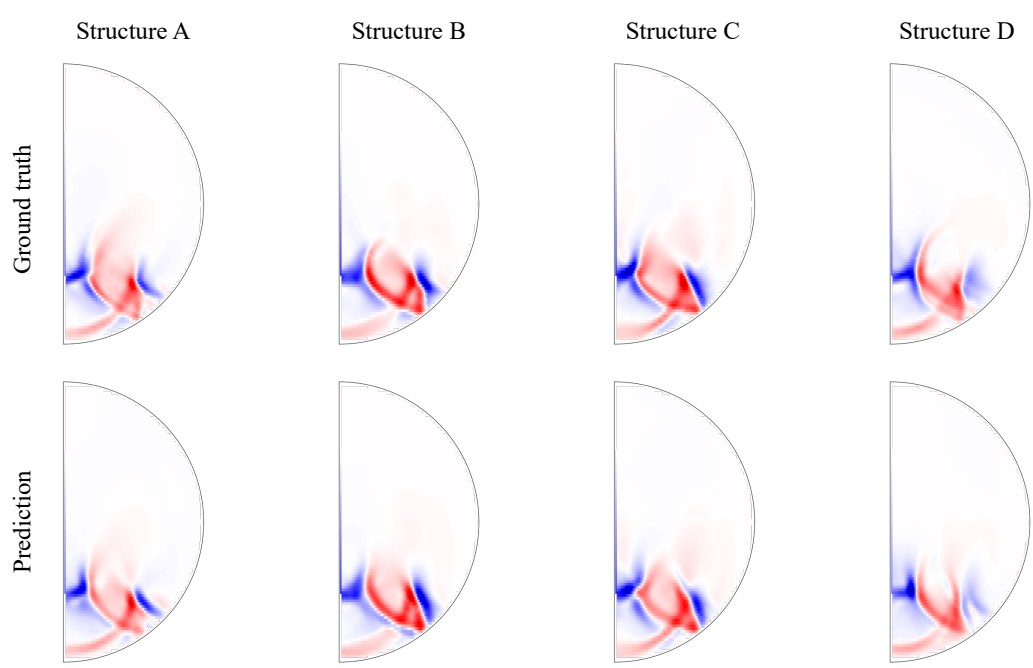

Figure A4: **Snapshot 11 of forward modeling.** We show the ground truth and predicted wavefield snapshot within four velocity structures: A, B, C, and D.

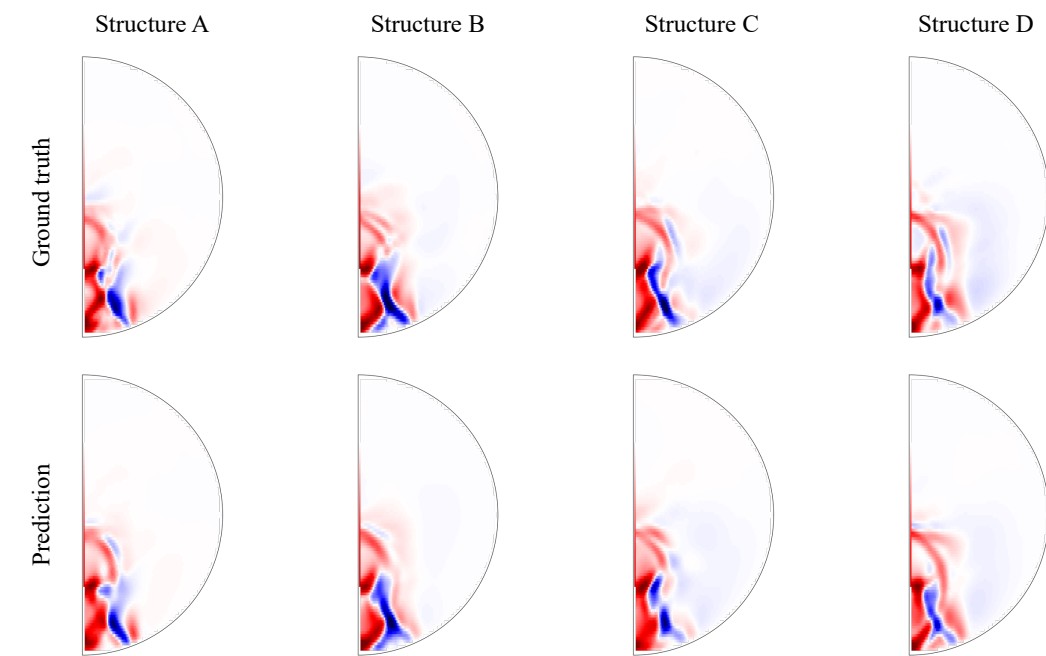

Figure A5: **Snapshot 13 of forward modeling.** We show the ground truth and predicted wavefield snapshot within four velocity structures: A, B, C, and D.

## H.2  INVERSION

We present additional visualization of inversion results using various unseen seismogram data. The figures display both the actual and the inverted velocity perturbations across five depths, along with their correlations. Refer to figs. A6 to A15.

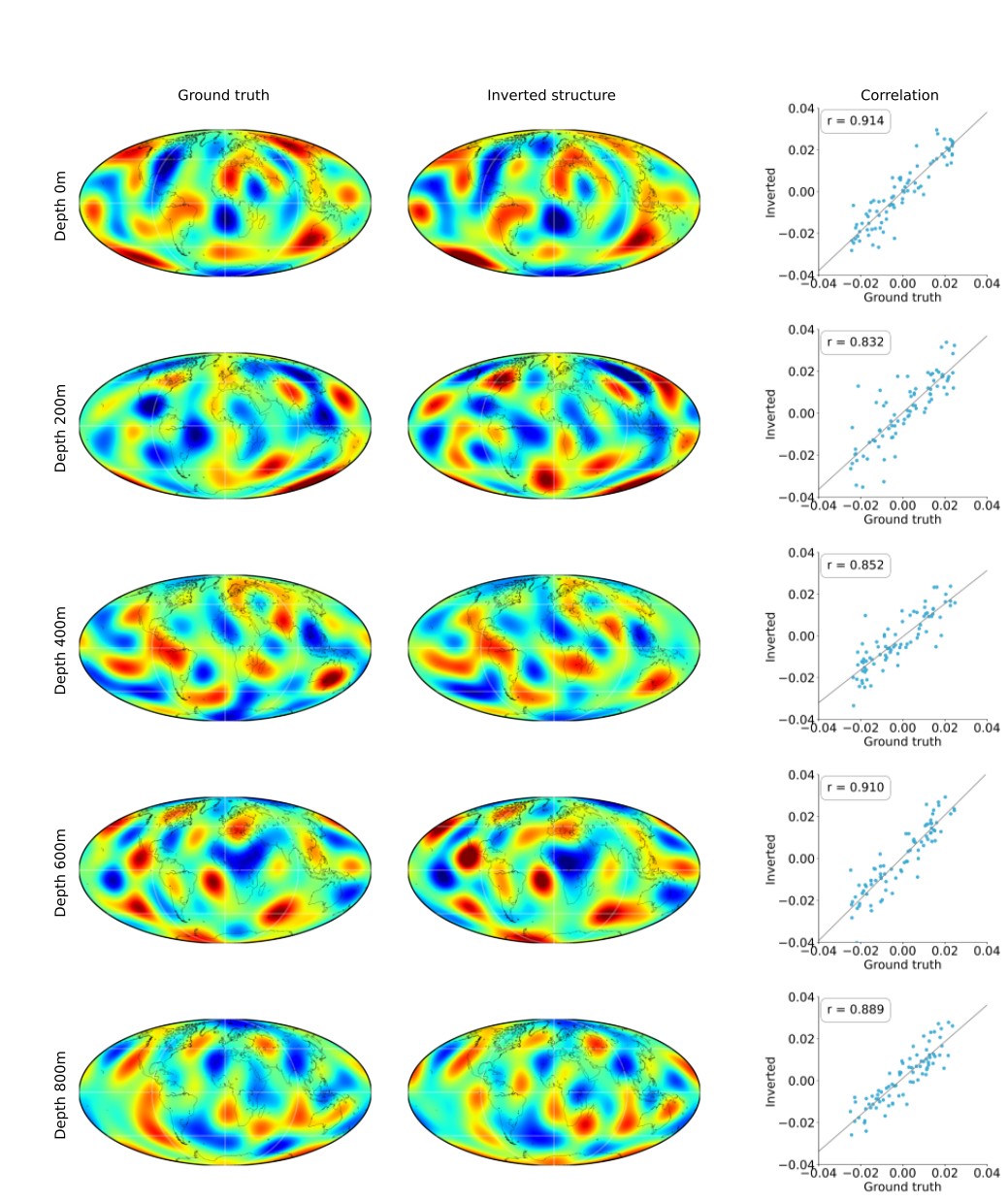

Figure A6: **Example 1 of acoustic inversion.** The colors range from blue to red, representing velocity perturbations from negative to positive. There are five rows, each depicting the velocity structure at different depths: 0m, 200m, 400m, 600m, and 800m. The first column displays the actual velocity structure, the second column shows the structure as derived from direct inverse mapping, and the third column illustrates the correlation between the spherical harmonics of the actual structure and those of the inverted structure.

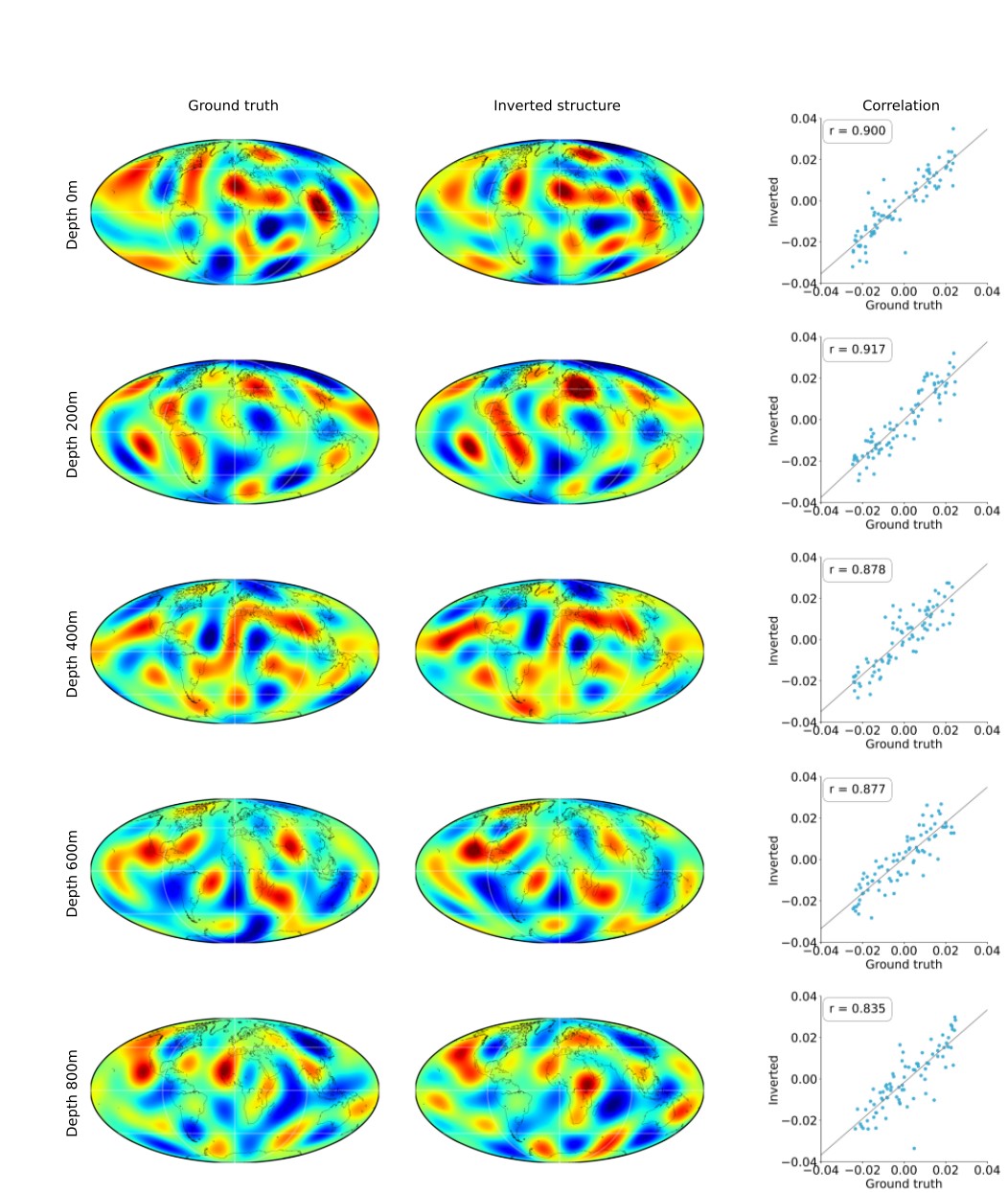

Figure A7: **Example 2 of acoustic inversion.** The colors range from blue to red, representing veloc-ity perturbations from negative to positive. There are five rows, each depicting the velocity structure at different depths: 0m, 200m, 400m, 600m, and 800m. The first column displays the actual velocity structure, the second column shows the structure as derived from direct inverse mapping, and the third column illustrates the correlation between the spherical harmonics of the actual structure and those of the inverted structure.

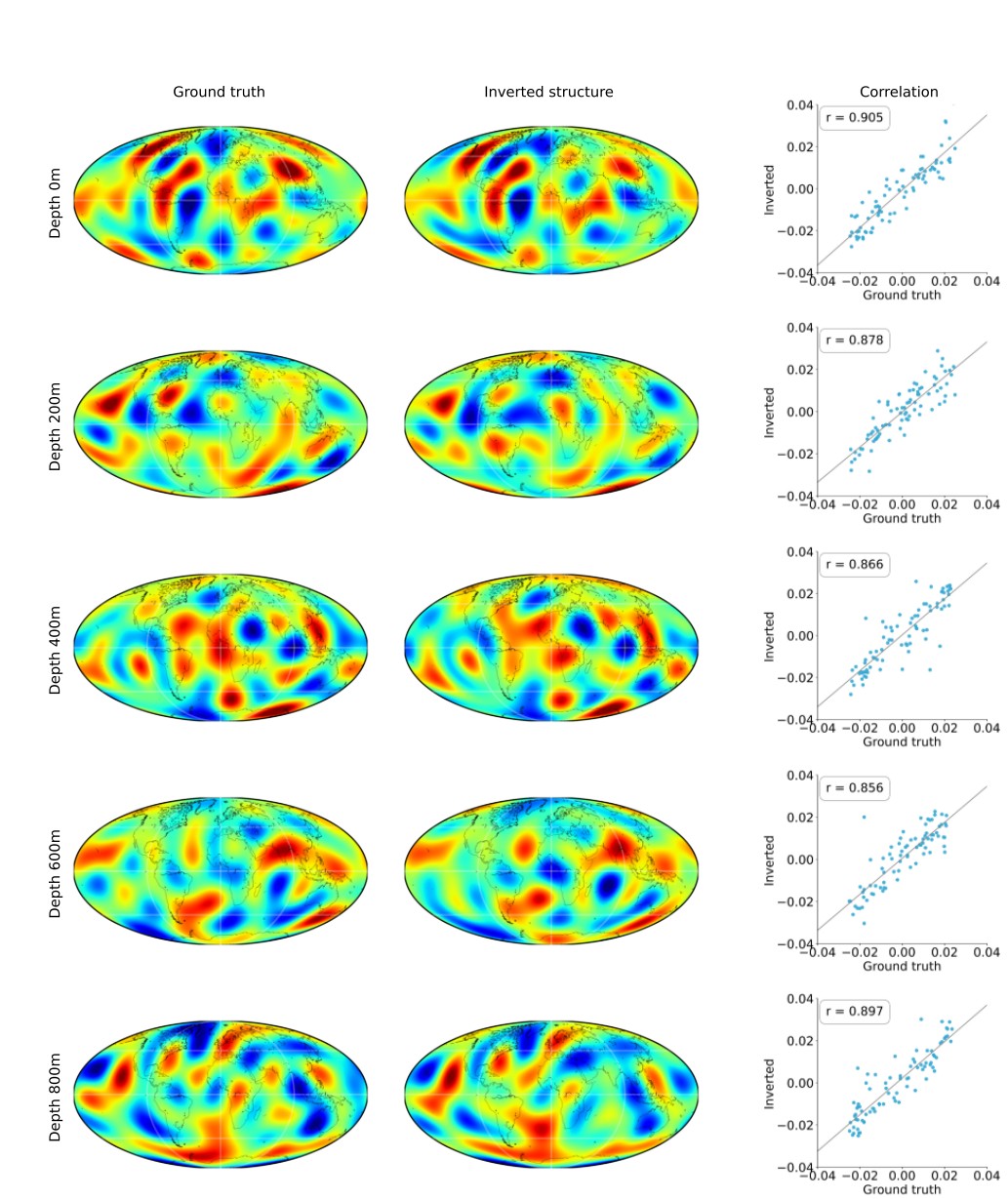

Figure A8: **Example 3 of acoustic inversion.** The colors range from blue to red, representing veloc-ity perturbations from negative to positive. There are five rows, each depicting the velocity structure at different depths: 0m, 200m, 400m, 600m, and 800m. The first column displays the actual velocity structure, the second column shows the structure as derived from direct inverse mapping, and the third column illustrates the correlation between the spherical harmonics of the actual structure and those of the inverted structure.

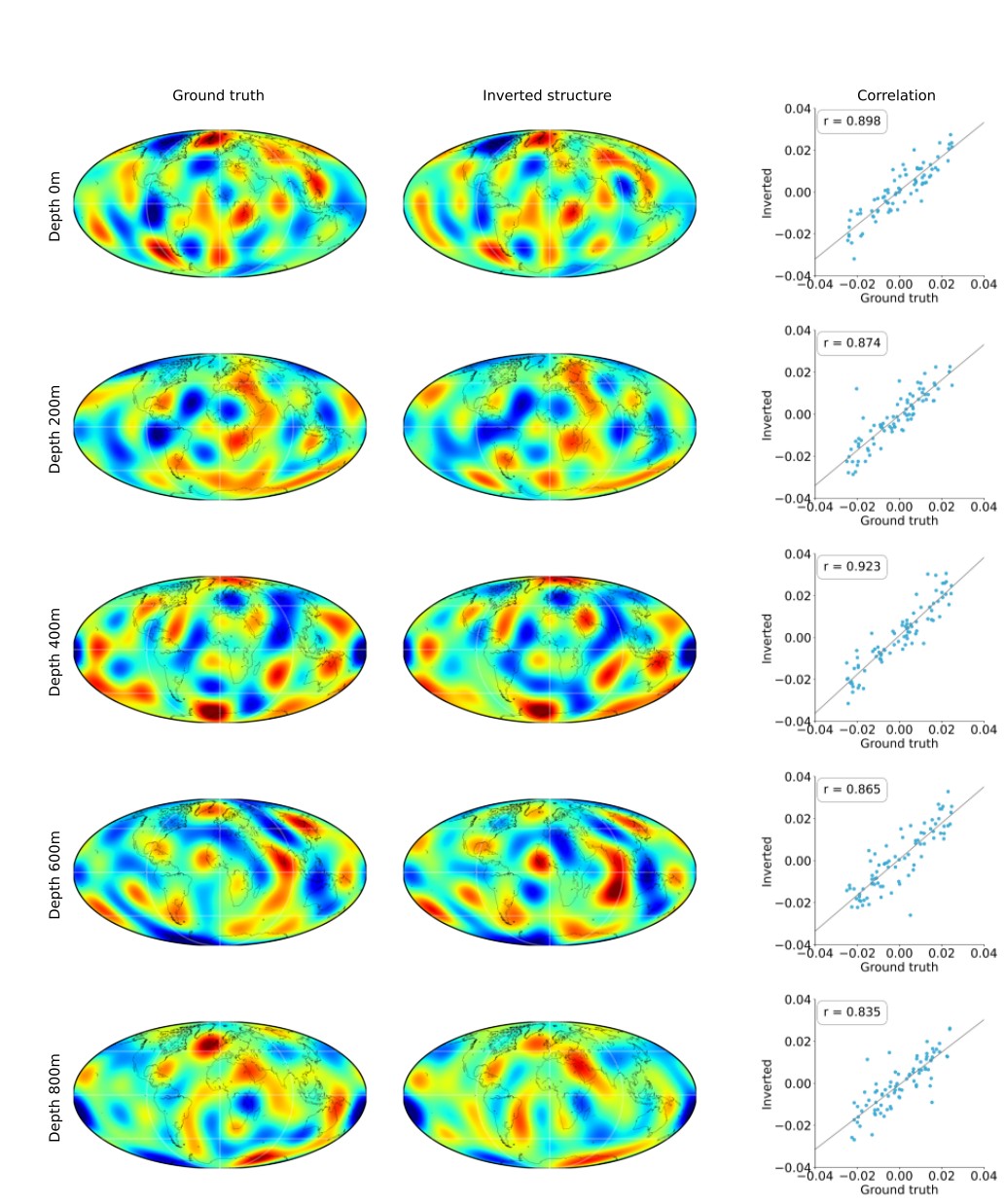

Figure A9: **Example 4 of acoustic inversion.** The colors range from blue to red, representing velocity perturbations from negative to positive. There are five rows, each depicting the velocity structure at different depths: 0m, 200m, 400m, 600m, and 800m. The first column displays the actual velocity structure, the second column shows the structure as derived from direct inverse mapping, and the third column illustrates the correlation between the spherical harmonics of the actual structure and those of the inverted structure.

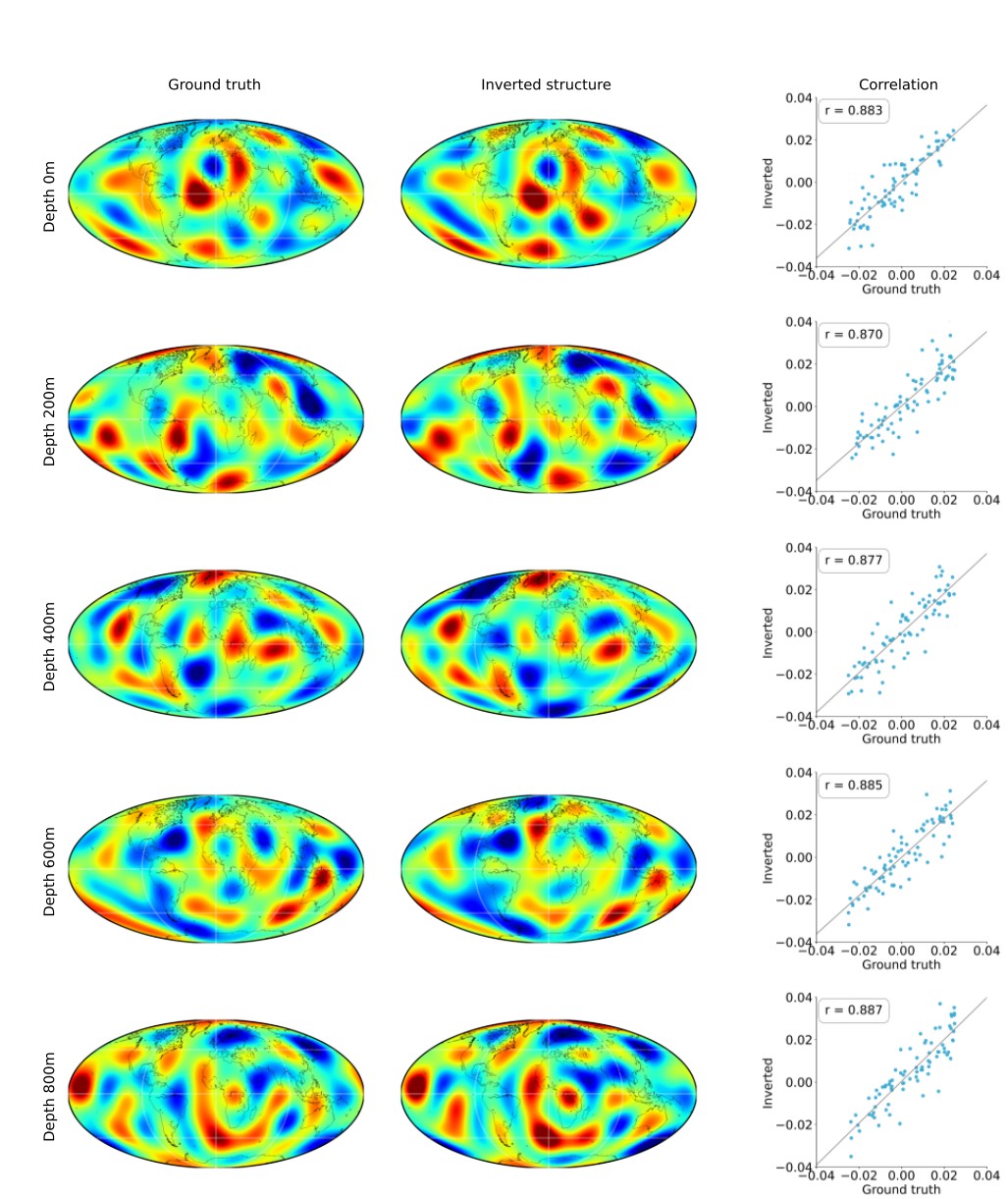

Figure A10: **Example 5 of acoustic inversion.** The colors range from blue to red, representing velocity perturbations from negative to positive. There are five rows, each depicting the velocity structure at different depths: 0m, 200m, 400m, 600m, and 800m. The first column displays the actual velocity structure, the second column shows the structure as derived from direct inverse mapping, and the third column illustrates the correlation between the spherical harmonics of the actual structure and those of the inverted structure.

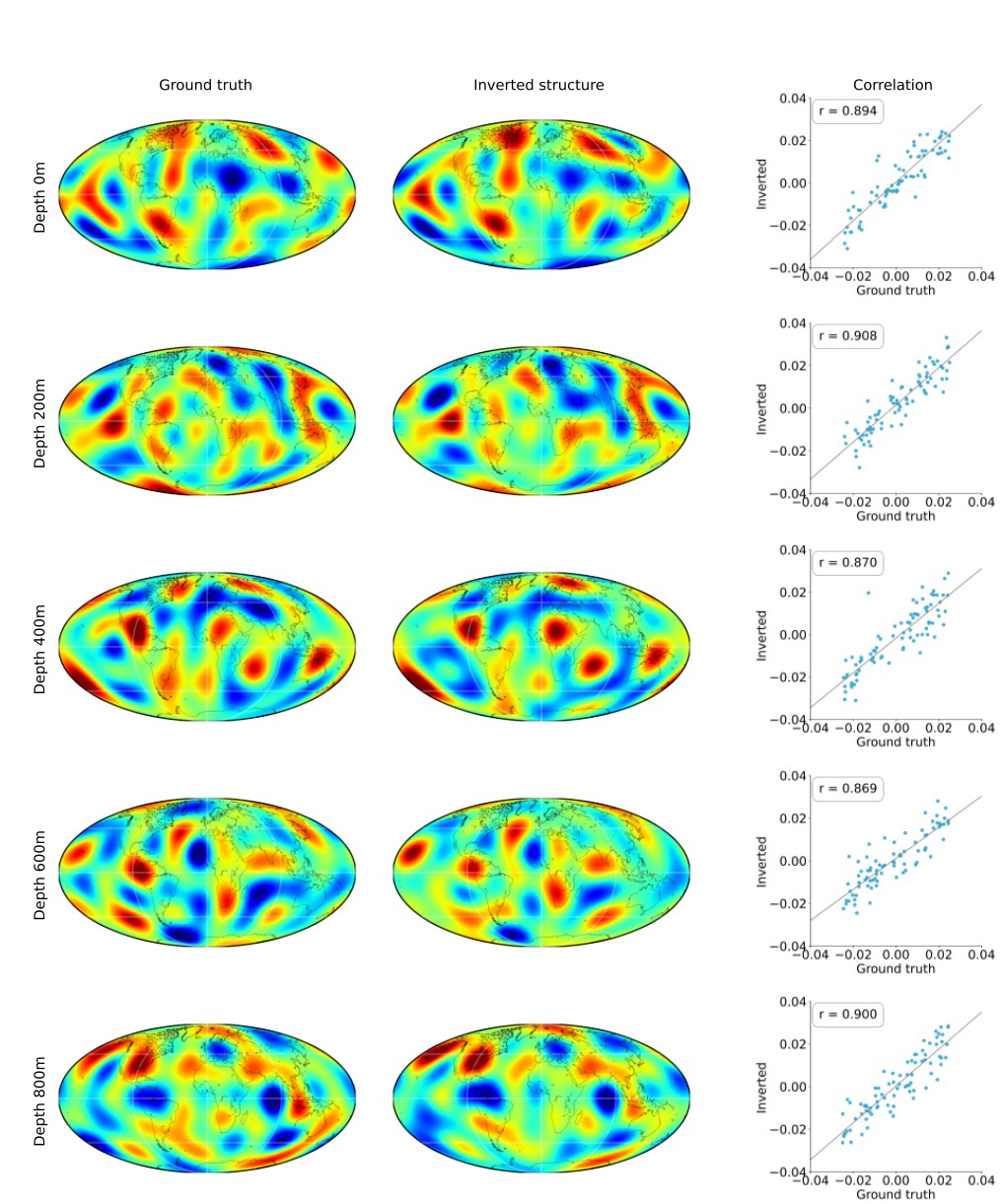

Figure A11: **Example 6 of acoustic inversion.** The colors range from blue to red, representing velocity perturbations from negative to positive. There are five rows, each depicting the velocity structure at different depths: 0m, 200m, 400m, 600m, and 800m. The first column displays the actual velocity structure, the second column shows the structure as derived from direct inverse mapping, and the third column illustrates the correlation between the spherical harmonics of the actual structure and those of the inverted structure.

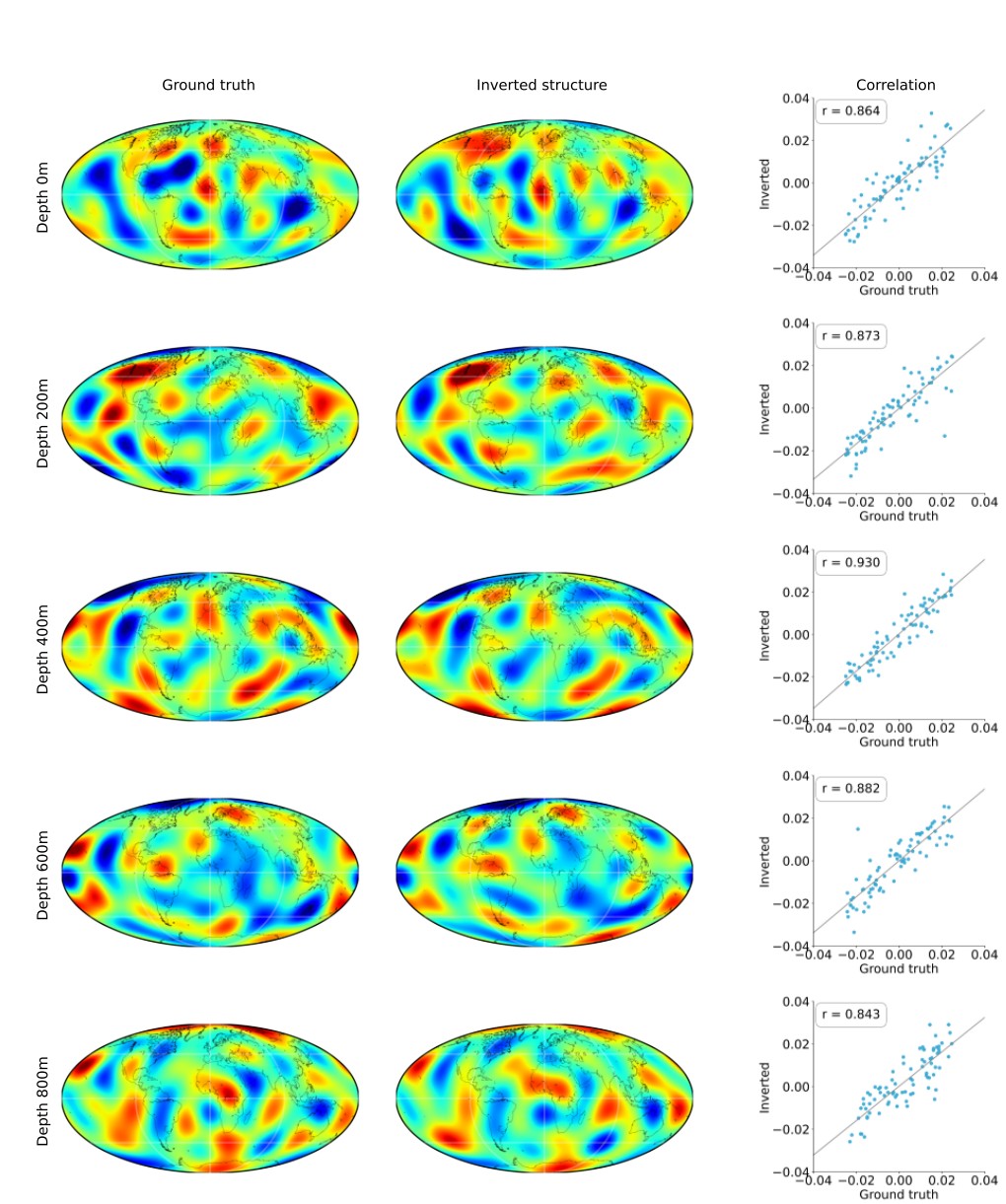

Figure A12: **Example 7 of acoustic inversion.** The colors range from blue to red, representing velocity perturbations from negative to positive. There are five rows, each depicting the velocity structure at different depths: 0m, 200m, 400m, 600m, and 800m. The first column displays the actual velocity structure, the second column shows the structure as derived from direct inverse mapping, and the third column illustrates the correlation between the spherical harmonics of the actual structure and those of the inverted structure.

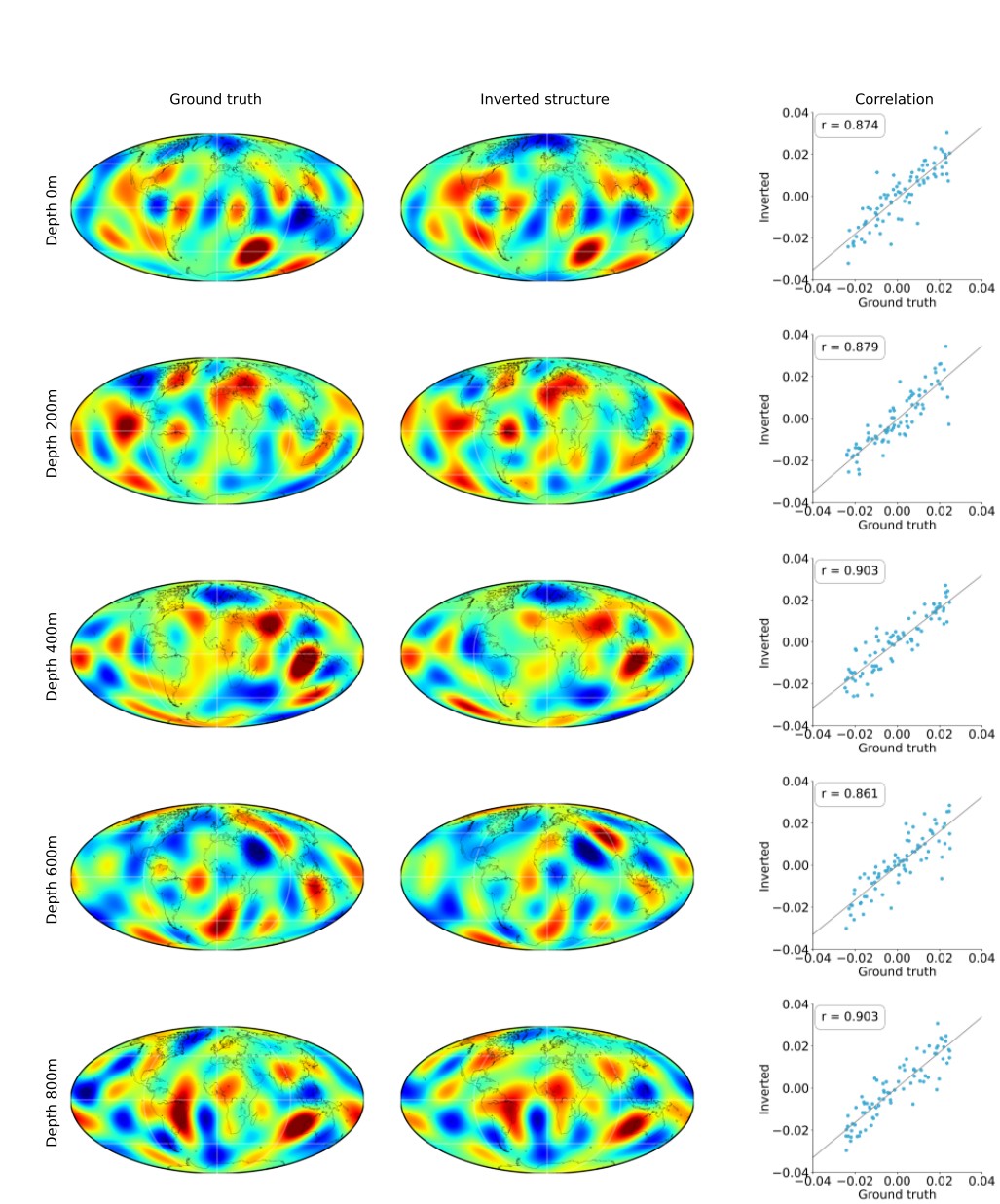

Figure A13: **Example 8 of acoustic inversion.** The colors range from blue to red, representing velocity perturbations from negative to positive. There are five rows, each depicting the velocity structure at different depths: 0m, 200m, 400m, 600m, and 800m. The first column displays the actual velocity structure, the second column shows the structure as derived from direct inverse mapping, and the third column illustrates the correlation between the spherical harmonics of the actual structure and those of the inverted structure.

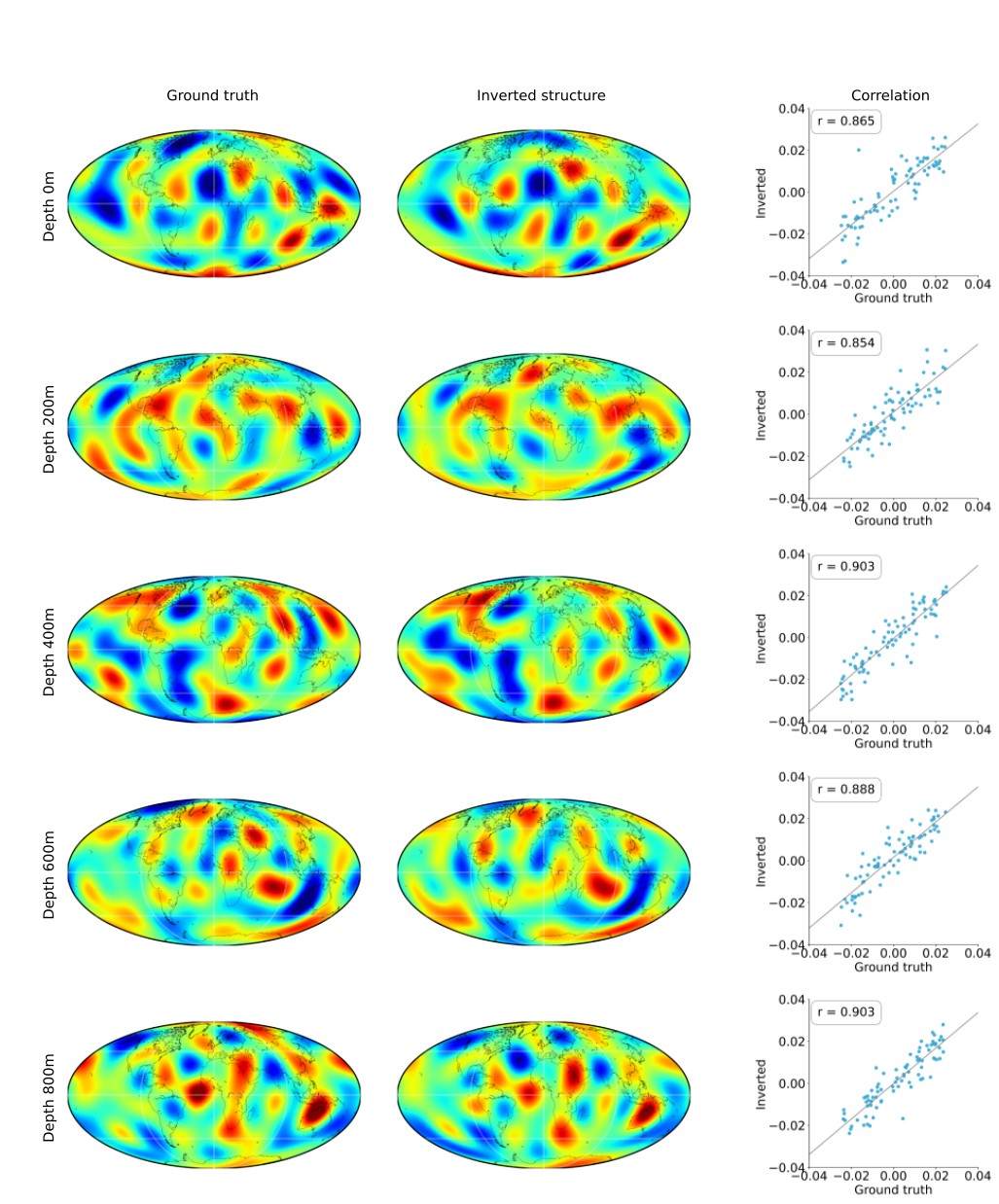

Figure A14: **Example 9 of acoustic inversion.** The colors range from blue to red, representing velocity perturbations from negative to positive. There are five rows, each depicting the velocity structure at different depths: 0m, 200m, 400m, 600m, and 800m. The first column displays the actual velocity structure, the second column shows the structure as derived from direct inverse mapping, and the third column illustrates the correlation between the spherical harmonics of the actual structure and those of the inverted structure.

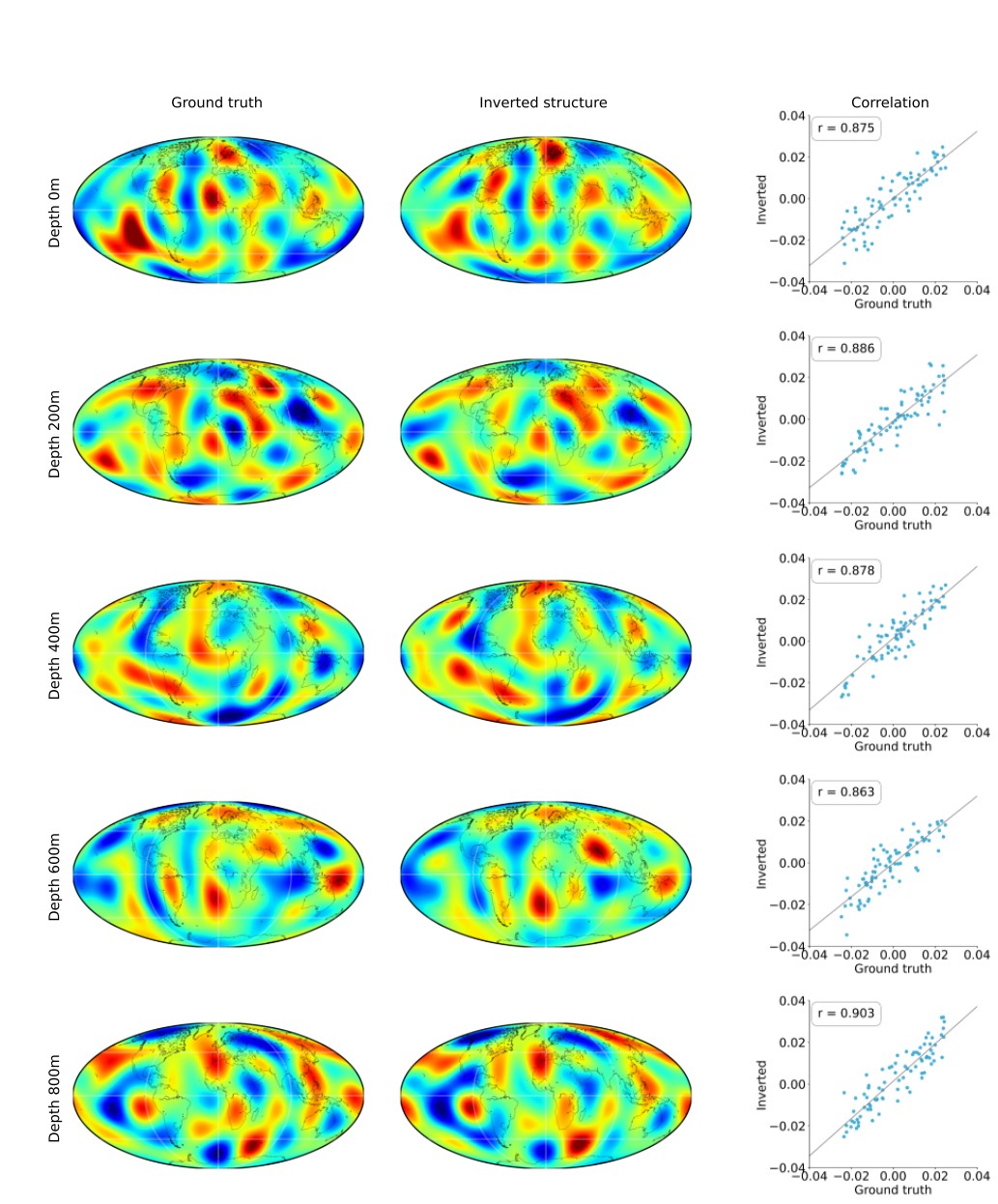

Figure A15: **Example 10 of acoustic inversion.** The colors range from blue to red, representing velocity perturbations from negative to positive. There are five rows, each depicting the velocity structure at different depths: 0m, 200m, 400m, 600m, and 800m. The first column displays the actual velocity structure, the second column shows the structure as derived from direct inverse mapping, and the third column illustrates the correlation between the spherical harmonics of the actual structure and those of the inverted structure.

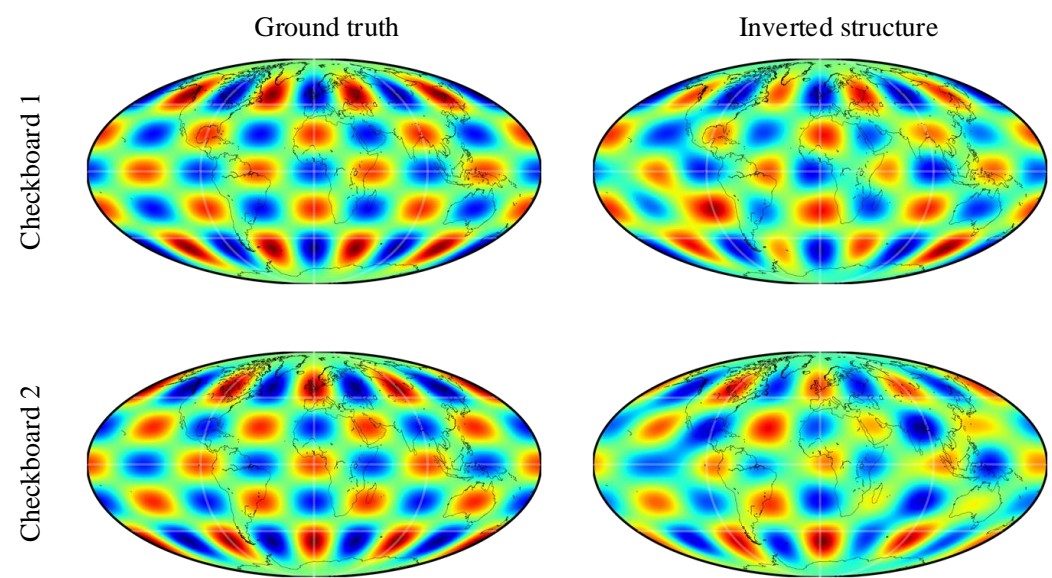

Figure A16: **A checkboard test on our ML-based inversion.** This test demonstrates the robustness and generalization of our inversion workflow.

### H.3    A CHECKBOARD TEST

To validate the robustness of our inversion methods, we conducted a checkerboard test as a standard diagnostic tool in geophysical tomography (Rawlinson et al., 2014; Tromp, 2020). The test involves simulating a known subsurface model with alternating high and low-velocity regions arranged in a checkerboard pattern, which allows us to evaluate the algorithm's ability to recover fine-scale varia-tions in the subsurface. We evaluate the checkboard inversion using our pre-trained InversionMLP. The inversion results demonstrated a high correlation of 0.89 with the ground truth structures, with the algorithm successfully resolving the checkerboard structure and accurately recovering the mod-eled velocity contrasts; see fig. A16 for visualization. This robust performance indicates that our approach is capable of handling complex geophysical inversion tasks and offers significant potential for real-world applications in seismic tomography and other geophysical imaging techniques.

