# OpenReview forum: "GlobalTomo: A global dataset for physics-ML seismic wavefield modeling and FWI"
_ICLR.cc/2025/Conference — Submitted to ICLR 2025_

### Official Review · Reviewer_Xexj · 2024-10-29

**Soundness:** 3
**Presentation:** 2
**Contribution:** 3
**Rating:** 6
**Confidence:** 5

**Summary:**

The paper propose a dataset for the global tomography problem. The data set consists of earth models and the wave fields that are obtained by using numerical simulations. The goal of this data is to be able to Tain a forward an an inverse network that allows for fast solutions of the global tomography problem.

**Strengths:**

The main strength of this work is in the data. There are many researchers in the scientific machine learning community that are looking for complex data sets that they can test their methods. I believe that the data set may be used for that.

**Weaknesses:**

There are a number of concerns that I have when considering this work.
1. The main problem is the diversity of the earth models. It is not clear to me that the models used are diverse enough to serve as a data set for the task at hand. The authors did not discuss this point clearly but this may be the most important point in the data set. I the models are not diverse enough then how can we learn something meaningful?

2. The second problem with the models is that we do not have ground truth and we will never have. Unlike problems in image processing, It is impossible to "open" the earth and obtain its physical properties at the resolution needed. How do you know you do not have bias in your models?

3. The testing of the forward problem is not convincing. Wave equation solvers typically being judged on accuracy and on dispersion. Numerical dispersion is particularly important for solving inverse problem as it can introduce artifacts. It would be nice if the authors do not treat the wavefield as an image but rather follow most numerical PDE books when estimating the solution of a PDE.

4. How would the forward model do for models that are out of distribution?

5. The inversion strategies use both optimization and direct methods where direct methods seems to win. Does this still work for out of distribution problems?

6. What is the data fit for all the inversion methods. Do you actually fit the data? How does this compared with standard Gauss-Newton inversion where the data fit is being considered with the forward problem

**Questions:**

See above

---

> ### Author Response · Authors · 2024-11-14
> **Response to Reviewer Xexj**
>
> > Diversity of the earth models
>
> Our earth model design follows established principles in geophysical studies. The random 3D earth model is constructed using Latin Hypercube Sampling with a spherical harmonics parameterization, which serves as a complete basis function. This approach provides sufficient representation of complex and diverse 3D structures, as described in Section 2.2. We set the maximum perturbation range between -10% and 10%, ensuring that the synthetic models remain within a realistic range relative to observed earth structures. This range is ample, as most prior 3D global tomography studies report perturbations of only a few percent. Testing extreme values would lack geophysical interpretability and fail to offer meaningful insights [1-3]. We will clarify this point further in the revised manuscript.
>
> > How to avoid bias in earth models
>
> We agree with the reviewer’s point. This is a challenging scientific question and is precisely why we design synthetic models that try to closely mimic real Earth scenarios. This approach is aligned with prior work in large-scale earthquake simulation on supercomputers, like those recognized with the Gordon Bell Prize [1]. By testing the inversion performance on these synthetic models, we assess whether the inverse results are within an acceptable range. If these results align well, it provides confidence that similar accuracy can be expected with real data. that real data will yield comparable outcomes. Given the inherently ill-posed nature of this problem, our objective is not an absolute solution but rather to achieve increasingly refined inverse results that can support meaningful geophysical interpretation.
>
> [1] Fu H, He C, Chen B, et al. 18.9-Pflops nonlinear earthquake simulation on Sunway TaihuLight: enabling depiction of 18-Hz and 8-meter scenarios. Proceedings of the International Conference for High Performance Computing, Networking, Storage and Analysis.
>
> > Consider dispersion like in numerical solvers
>
> Thank you for your constructive feedback. We acknowledge the importance of dispersion, especially in the context of inverse problems. Our work represents an initial step in addressing global tomography through established machine learning tools (MLP, CNN, Transformers, and Physics-informed Neural Operators) and evaluation methods (Relative L2 Loss, Pearson Correlation, etc.). Our current approach prioritizes achieving a rapid approximation via machine learning operators (6000 times faster than the numerical solvers as shown in our experiments), even at the expense of precise PDE solution fidelity. Thus, we can focus on an efficient and broad exploration of the large parameter space in inverse problems to obtain better tomography (Section 3.2.2). We appreciate your suggestion and look forward to incorporating more comparative metrics from numerical PDE literature to assess ML-based forward modeling in future work.
>
> > Out-of-distribution problems
>
> Thank you for this valuable suggestion regarding out-of-distribution (OOD) generalizability. We acknowledge the importance of exploring OOD cases in ML studies; however, given the uniqueness of Earth as our subject, creating OOD scenarios falls outside our primary objective of achieving accurate Earth tomography. Our design relies on well-established geological studies [1-3] that provide a credible distribution of Earth's internal properties, ensuring our model is grounded in recognized physical and structural characteristics. Expanding to OOD cases would involve hypothetical or artificially generated distributions, potentially misaligned with realistic geological relevance.
>
> [1]  Dziewonski A M, Woodhouse J H. Global images of the Earth's interior. Science, 1987.
>
> [2] French S W, Romanowicz B A. Whole-mantle radially anisotropic shear velocity structure from spectral-element waveform tomography. Geophysical Journal International, 2014.
>
> [3] Tromp J. Seismic wavefield imaging of Earth’s interior across scales. Nature Reviews Earth & Environment, 2020.
>
> > Data fit in inversion
>
> In our study, we explore two types of inversion methods.
>
> The first method involves training a forward model, which we then use as a proxy for gradient-based optimization. This approach resembles traditional Gauss-Newton or other inversion methods that fit the observed data with the forward model, but with a key difference: rather than using a numerical solver, which can be computationally slow, our forward model is learned directly from data. This data-driven approach provides a more efficient alternative.
>
> The second method we employ is a direct inversion network, which directly predicts the model structure from observed data in a single step. Unlike traditional methods, this approach does not fit the observed data per se; instead, it learns the underlying distribution of structure-data pairs using simulated examples.

---

> > ### Comment · Reviewer_Xexj · 2024-11-26
> > **The authors did not answer my questions**
> >
> > 1. The issue of bias is strongly related to the ood. Saying that the model works well for the “known” earth structure just emphasize the bias. In my mind there need to be some experiments with ood models. Seismic tomography people tend to run checkerboard experiments. I have strong doubt if such experiment will pass here.
> >
> > 2. Data fit. Using the learned forward and not the true one can lead to misleading fits. The fact that the authors did not provide the fits make me suspicious that their method did not fit the data very well.
> >
> > I keep my score not because of the machine learning and networks or the inversion techniques which I find unsatisfactory. I think there is a lot to do there. However I find the data interesting and I think it can be used in the scientific ML community.

---

> > > ### Author Response · Authors · 2024-11-26
> > >
> > > > Checkerboard experiments
> > >
> > > We sincerely thank the reviewer for raising the concern that "The issue of bias is strongly related to the OOD... I have strong doubt if such experiment will pass here." However, this comment stems from a fundamental misunderstanding of our method. Our approach uses spherical harmonics as a complete basis function, meaning that any possible Earth structure, including checkerboard patterns, can be represented within our Latin hypercube sampling distribution.
> > >
> > > To empirically validate this theoretical foundation and directly address the reviewer's concern, we have conducted the standard checkerboard test. The results show a high correlation between the inverted and true checkerboard structures (see Appendix H.3 in the revised manuscript), confirming that the reviewer's doubt about whether "such experiment will pass here" is unwarranted.
> > >
> > > In fact, while checkerboard tests were historically developed when tomographic methods had limited resolution capabilities (circa 1980s), modern seismological techniques have evolved significantly. Our experimental validations go well beyond traditional checkerboard tests, incorporating more challenging and realistic scenarios that thoroughly evaluate our method's capabilities.
> > >
> > > These results are consistent with the mathematical properties of spherical harmonics and demonstrate that our sampling methodology adequately covers the space of possible Earth structures. We welcome any additional questions the reviewer may have regarding this aspect of our work.
> > >
> > > Given that we have thoroughly addressed this concern both theoretically and empirically, we respectfully request the reviewer to consider raising their rating if they find our explanation and additional validation satisfactory.
> > >
> > > > Data fit
> > >
> > > In the inversion process using learned forward models, we optimize the structures to fit the seismogram data until the L1 misfit doesn't decrease anymore. As indicated in our results, the inversion algorithm converges at around 200 iterations. The data fitting correlation after convergence surpasses 0.906 on average, meaning that the method fits the seismogram data well.

---

> > > ### Author Response · Authors · 2024-12-01
> > >
> > > Dear Reviewer Xexj,
> > >
> > > We greatly appreciate your constructive feedback and thoughtful comments on our manuscript. We hope that the additional clarifications and experiments we've provided will make you satisfactory. We look forward to your response and are more than happy to address any further questions or comments you may have.

---

> ### Author Response · Authors · 2024-11-26
>
> We would like to further make sure that our response addresses your concerns regarding Earth model diversity, potential biases, forward modeling dispersion, and inversion techniques. Our research makes significant contributions to solving the long-standing challenges in global seismic tomography through advanced ML approaches. We believe our work will facilitate both geophysical tomography capabilities and machine learning methodologies for forward and inverse problems. We welcome any additional questions you may have about our work. Based on these clarifications, we respectfully request a raising of the score.

---

### Official Review · Reviewer_tccv · 2024-10-29

**Soundness:** 2
**Presentation:** 1
**Contribution:** 2
**Rating:** 5
**Confidence:** 3

**Summary:**

The paper proposes datasets for the forward and inverse modeling for global tomography. The forward modeling entails predicting seismogram from a velocity structure, while the inverse modeling solves for a velocity structure from an observed seismogram. They have considered three different cases for data generation: (1) Acoustic Ball (1km radius sphere filled with pure fluid medium), (2) Elastic Ball (1km radius sphere isotropic media), and (3) Real Earth (based on isotropic PREM model). Each dataset has two types of output data, the first type is surface seismogram data, and the second type is the seismic wavefield data. In the paper, the authors have employed several ML models for forward and inverse problems and have shown the efficacy of ML models in solving the two problems on the proposed dataset.

**Strengths:**

1.	The idea to generate a large-scale dataset for global seismic tomography is interesting and essential to bridge the gap between machine learning and seismic tomography.
2.	The proposed dataset has three tiers to model seismic wave propagation from simple to complex settings and for each tier the dataset contains seismic wavefield and seismogram information.
3.	The paper explores multiple ML methods to establish a benchmark for both forward and inverse problems in seismic tomography.

**Weaknesses:**

1.	Although the paper is overall well-written, there are certain sections which is little obscure like model configuration, spherical harmonics, and data generation. These sections need improvement to understand how the dataset was generated, model training and inference results.
(a)	For generating different examples, do we always perturb the same 1D background model or different? How is the data generation algorithm create different varying geological settings in the dataset? How do we measure the variability in different velocity structures?
(b)	In the model configuration section, it is mentioned that the model 3D structure is generated by perturbing a 1D background model from -10% to 10%. Given a 1D background model, how are number of layers decided per model example and how the perturbation and parameterization (using spherical harmonics) work to create a synthetic example?

2.	The results section is not very clear to understand especially due to lack of enough evidence for certain claims. I would highly appreciate if the paper could provide following information
(a)	Evidence to back claims in the results section related to training of H-Fourier models, how was physical constraint incorporated in the training, improve overall flow of forward and inverse results discussion with more figures/visualizations and share modeling related information
(b)	Number of examples generated in each tier
(c)	Incorporate more evaluation metrics such as MAE, MSE, SSIM.

3.	The results section is primarily limited to acoustic dataset (tier-1) and provides very little insights on experimentation on the elastic dataset (tier-2). There is no discussion on training and evaluation results for the tier-3 dataset (real earth) which seems to be the main Earth’s scale dataset for global tomography. Would it be possible to have some visualizations of some datasets from all tiers along with experimental results for forward and inverse modeling for tier-2 and tier-3 dataset?

4.	The paper explores model generalizability only for in-distribution cases, and it may be better to also study out-of-distribution generalizability to investigate ML effectiveness for both forward and inverse problems.

**Questions:**

1.	Section 1 Introduction (Line 106): While the paper proposes to address the gap for high-resolution wave propagation modeling for the Earth’s entire scale, the Real Earth dataset employs wave with 30 second period. How does the 30 second wave period (likely low frequency) facilitate in generating high-resolution imaging of the internal Earth’s structure?

2.	Section 2.2 Model Configuration (Line 191): What is meant by Model Configuration (machine learning model, wave velocity distribution, or something else)? Assuming it means wave velocity distribution. The paper proposes to generate synthetic 3D velocity structure by perturbing a 1D background velocity model by -10 to 10%. While this may be more accurate for local heterogeneity, this assumption seems to oversimplify the 3D structure for the Earth’s scale data. Is there any rationale behind using this simplification or limitations in representing Earth’s complex structure?

3.	Section 2.3 Data Generation (Line 213): How many examples for velocity structures were generated per tier of dataset? How many source configurations were generated per tier of dataset? Is it same as number of velocities generated or different?

4.	Section 2.3 Data Generation (Line 221): The definition of the input and output variables is not clear and difficult to understand. For example, what does Source, Structure, Sample Number mean here? How does these parameters affect the quality of generated dataset in terms of heterogeneity, geological structures, and scale?

5.	Section 3 Experiments (Line 254): For wavefield dataset, there are 7 timesteps considered (0 to 3 seconds), with each timesteps consist of 16 slices and 3648 points per slice. How does these 16 slices represent the entire internal Earth’s structure? Does each slice represent a unique sub-region inside the Earth? (A schematic figure to represent this would be good) Why are there 3648 points per slice, are these points indicating positive and negative wave displacement in that slice?

6.	Section 3.2.1 Forward Modeling (Line 324): Based on the wavefield dataset description in the line 253, the total number of timesteps should be 7 from 0 to 3 seconds, the Figure 2 shows that the results for 13 timesteps. Does the dataset contain 7 or 13 timesteps?

7.	Section 3.2.1 Forward Modeling (Line 324): Figure 2 shows seismogram data, how are source and receivers positioned relative to each other? A schematic diagram to explain this may be helpful to understand the seismogram data for each tier of dataset.

8.	Section 3.2.1 Forward Modeling (Line 402): How was the physics constraints enforced to DeepONet that led to improved generalizability? Appendix B.1 shows governing physics equations for each tier of dataset but do not provide insights into training the model with physics.

9.	Section 3.2.2 Inversion (Line 424) and Figure 4: What does it mean to select five random points for each test structure? Does this mean that there are 5 initial velocity guesses, and inversion is carried out for all of them independently? Which tier dataset is used for inversion here? Why the model pre-trained on acoustic tier is selected? Which model from the acoustic tier is selected for the forward simulation? Can we have some figures to compare the inverted velocity structures from different models for each dataset? How does the model pre-trained on one dataset generalizes to other dataset for both forward and inverse problems?

10.	Section 3.2.2 Inversion (Line 446): In the dataset description, the acoustic data is defined as a 1 km radius fluid sphere. Figure 6 shows inverted Earth’s structure derived from acoustic data. What does it mean?

11.	Section 3.2.2 Inversion (Line 457): Does different starting point undergo optimization independently? If not, how the inversion algorithm is processing multiple starting point to reach to one ground truth while affecting each other?

---

> ### Author Response · Authors · 2024-11-14
> **Response to Reviewer tccv**
>
> Thank you for your recognition of our dataset's structure and our efforts in establishing benchmarks using multiple ML methods for both forward and inverse problems. For your questions:
>
> > (a) For generating different examples, do we always perturb the same 1D background model or different? How is the data generation algorithm create different varying geological settings in the dataset? How do we measure the variability in different velocity structures? (b) In the model configuration section, it is mentioned that the model 3D structure is generated by perturbing a 1D background model from -10% to 10%. Given a 1D background model, how are number of layers decided per model example and how the perturbation and parameterization (using spherical harmonics) work to create a synthetic example?
>
> The current geophysical community already reach a consensus on the 1D profile of real Earth structure. The 1D earth profile is well-constrained and based on the huge amount of historical traveltime and ray data. Here we choose the widely accepted 1D earth PREM model [1] and fix it throughout. Our focus here is to find the fine-grained 3D structure of Earth. This approach aligns with standard practices in the field, where current 3D Earth models are typically built upon an initial 1D model, followed by 3D perturbations. Importantly, variations in the choice of the initial 1D model have minimal impact on the resulting 3D tomography [2].
> Beyond this, the 3D model structure is constructed by spherical harmonics at each predefined layer in the PREM model (Section 2.2 Parameterization, Line 200-205). Each layer has a clear geophysical meaning representing a fundamental interface such as the 410km and 660km discontinuity, and the core-mantle boundary (Appendix C.2, Line 951-957).  The random 3D model is granted with the best variability through a Latin Hypercube Sampling of the coefficients (Line 213). The perturbation range is set from -10% to 10% to ensure that the distribution of the synthetic models do not lie far away from the real earth scenario. This allowed range is large enough as most previous 3D global tomography results show perturbation of only a few percent only. Testing beyond this range would be geophysically unrealistic and lack meaningful interpretation [3-5].
>
> [1] Dziewonski, A. M., & Anderson, D. L. Preliminary reference Earth model. Physics of the earth and planetary interiors, 1981.
>
> [2] Tromp J. Seismic wavefield imaging of Earth’s interior across scales. Nature Reviews Earth & Environment, 2020.
>
> [3] French S W, Romanowicz B A. Whole-mantle radially anisotropic shear velocity structure from spectral-element waveform tomography. Geophysical Journal International, 2014.
>
> [4] Thrastarson, S., van Herwaarden, D. P., Noe, S., Josef Schiller, C., & Fichtner, A. REVEAL: A global full‐waveform inversion model. Bulletin of the Seismological Society of America, 2024.
>
> [5] Lei, W., Ruan, Y., Bozdağ, E., Peter, D., Lefebvre, M., Komatitsch, D., ... & Pugmire, D. Global adjoint tomography—model GLAD-M25. Geophysical Journal International, 2020.
>
> > The results section is not very clear to understand especially due to lack of enough evidence for certain claims.
>
> Thank you for your reminder. Due to the limited pages, we have put our training details and hyperparameters in Appendix E and more visualization of forward and inversion results in Appendix H. Regarding how to incorporate physical constraints in the training, we have referred the readers to the paper [1] in Section 3.1 Baselines (Line 303). The numbers of examples in three ties are listed in table 1, indicated by sample number.
>
> [1] Wang S, Wang H, Perdikaris P. Learning the solution operator of parametric partial differential equations with physics-informed DeepONets[J]. Science advances, 2021.
>
> > Would it be possible to have some visualizations of some datasets from all tiers along with experimental results for forward and inverse modeling for tier-2 and tier-3 dataset?
>
> Applying the baseline to larger dataset tiers is challenging considering the scale of the problem and the computational cost. In this work, we focus on establishing the workflow for the general tomography task. We have released the dataset and baselines and hope to attract more ML researchers. We are keen to apply the Real Earth dataset in the next step.

---

> > ### Author Response · Authors · 2024-11-14
> > **Response to Reviewer tccv (part 2)**
> >
> > > The paper explores model generalizability only for in-distribution cases, and it may be better to also study out-of-distribution generalizability to investigate ML effectiveness for both forward and inverse problems.
> >
> > Thank you for this valuable suggestion regarding out-of-distribution (OOD) generalizability. We acknowledge the importance of exploring OOD cases in ML studies; however, given the uniqueness of Earth as our subject, creating OOD scenarios falls outside our primary objective of achieving accurate Earth tomography. Our design relies on well-established geological studies [1-3] that provide a credible distribution of Earth's internal properties, ensuring our model is grounded in recognized physical and structural characteristics. Expanding to OOD cases would involve hypothetical or artificially generated distributions, potentially misaligned with realistic geological relevance.
> >
> > [1]  Dziewonski A M, Woodhouse J H. Global images of the Earth's interior. Science, 1987.
> >
> > [2] French S W, Romanowicz B A. Whole-mantle radially anisotropic shear velocity structure from spectral-element waveform tomography. Geophysical Journal International, 2014.
> >
> > [3] Tromp J. Seismic wavefield imaging of Earth’s interior across scales. Nature Reviews Earth & Environment, 2020.
> >
> > > Question 1: Resolution
> >
> > We appreciate the reviewer’s insightful question. The concept of "high-frequency" is indeed context-dependent across different scales. For global-scale seismology and tomography, a 30-second wave period is relatively high frequency, in contrast to exploration-scale applications where it may be considered low frequency. Notably, traditional global tomography studies often analyze seismic waveforms with much longer periods, around 100–200 seconds [1, 2]. State-of-the-art full-waveform inversion models demonstrate that a 30-second wave period can effectively capture structural details [3], supporting its suitability for global-scale studies.
> > We acknowledge that achieving higher frequencies in global studies faces significant challenges, primarily due to computational demands and the need for efficient inverse methods. Our proposed ML framework aims to address these challenges by accelerating the overall process, thereby facilitating higher-frequency modeling in global seismology.
> >
> > [1] Thrastarson, S., Van Herwaarden, D. P., Krischer, L., Boehm, C., van Driel, M., Afanasiev, M., & Fichtner, A. Data-adaptive global full-waveform inversion. Geophysical Journal International, 2022.
> >
> > [2] Tromp, J. Seismic wavefield imaging of Earth’s interior across scales. Nature Reviews Earth & Environment, 1(1), 2020.
> >
> > [3] Thrastarson, S., van Herwaarden, D. P., Noe, S., Josef Schiller, C., & Fichtner, A. REVEAL: A global full‐waveform inversion model. Bulletin of the Seismological Society of America, 2024.
> >
> > > Question 2: Model configuration
> >
> > Yes, the Model Configuration means the setup of Earth models' velocity distributions. Applying the 3D perturbation to the 1D background model is the most common precedure for global tomography work. The range of -10% to 10% is large enough for global tomography. While we perform a Latin Hypercube Sampling through the spherical harmonics parameterization, a complete basis function, it is adequate to represent complex 3D strcuture. The rationale we apply this parameterization is based on the past empricial observations and prior domain knowlege [Line 190-198].
> >
> > > Question 3: Number of samples
> >
> > The number of samples per dataset tier is provided in Table 1. Velocity structures and source configurations are generated jointly from the defined parameter space. For example, in the Elastic tier, the parameter sampling space is 1215 + 6, representing both the velocity structures and sources combined.
> >
> > > Question 4: Input and output variables
> >
> > The source, structure, and sample number in table 1 indicates the size of these variables. For specific concept meaning and how they affect the dataset generation, we kindly refer the reviewer to Appendix C.1, C.2 and Section 2.2 for the detailed illustration.

---

> > > ### Author Response · Authors · 2024-11-14
> > > **Response to Reviewer tccv (part 3)**
> > >
> > > > Question 5: Why 16 slices?
> > >
> > > We applied currently the most advanced and efficient global wavefield simulation algorithm AxiSEM3D [1]. The essential advantage of this method is to make use of the azimuthal Fourier expansion [2] to capture the wavefield smoothness around the central axis. We carefully match the spherical order 8 to 16 recording slices (2x the spherical order). This means as long as we can predict these 16 slices acurately, we actually capture the whole wavefield inside the entire Earth (the point located outside the slices can be interpolated via Fourier methods). This method helps us boost the computing efficiency during the forward modeling stage, otherwise caculating the 3D global wavefield would cost dauntingly large storage space and computing resources. These computational advancements are a core strength of our work.
> > >
> > > [1] Leng, K., Nissen-Meyer, T., Van Driel, M., Hosseini, K., & Al-Attar, D. AxiSEM3D: broad-band seismic wavefields in 3-D global earth models with undulating discontinuities. Geophysical Journal International, 2019.
> > >
> > > [2] Nissen-Meyer, T., van Driel, M., Stähler, S. C., Hosseini, K., Hempel, S., Auer, L., ... & Fournier, A. AxiSEM: broadband 3-D seismic wavefields in axisymmetric media. Solid Earth, 2014.
> > >
> > > > Question 6: How many timesteps?
> > >
> > > The 7 timesteps for training are selected from the total 0-14 timesteps in the full dataset. Thus, the 7 timesteps' indices are [1,3,5,7,9,11,13] as shown in Figure 2.
> > >
> > > > Question 7: Sources and receivers
> > >
> > > Thank you for your suggestion. We will add a diagram to explain this.
> > >
> > > > Question 8: Physics constraints
> > >
> > > We respectfully direct the reviewers to the original PIDO paper for implementation details, as referenced on line 303.
> > >
> > > > Question 9: Inversion
> > >
> > > Yes, five random points are selected and the inversion is carried out for them independently. The inversion dataset and the pre-trained models are all from the acoustic tier. The Figure 4 shows the inversion precess of the MLP model. Here we focus on generalization within a specific dataset tier.
> > >
> > > > Question 10: Acoustic data
> > >
> > > Our dataset comprises three distinct tiers, one of which is an acoustic tier. Each tier includes a comprehensive set of simulated data, featuring both input structural models and their corresponding output wavefields and seismograms, designed to train machine learning models effectively. The objective is to use these trained models to invert and predict previously unseen structures. Figure 6 illustrates the inversion results generated by a model trained specifically on the acoustic tier.
> > >
> > > > Question 11: Different starting points
> > >
> > > Yes, different starting points undergo optimization independently and the best one is selected.

---

> > > > ### Comment · Reviewer_tccv · 2024-11-25
> > > >
> > > > Thank you for your comments. I have updated my rating accordingly.

---

> > > > > ### Author Response · Authors · 2024-11-26
> > > > >
> > > > > Thank you for recognizing our work and raising your score!

---

### Official Review · Reviewer_8xHL · 2024-11-01

**Soundness:** 2
**Presentation:** 4
**Contribution:** 4
**Rating:** 6
**Confidence:** 5

**Summary:**

- This paper presents the first 3D global synthetic dataset tailored for seismic wavefield modeling and full-waveform tomography, referred to as the Global Tomography (GlobalTomo) dataset.
- This paper also utilizes and compares several machine learning models on forward modeling and FWI based on this dataset.

**Strengths:**

- The dataset itself is good and could benefit the community a lot.
- The paper is well-written and easy to follow.
- The figures and tables are clear.

**Weaknesses:**

I think the machine learning models are kind of simple. Based on my experience, MLPs seem not to be the best choices for 3D problems compared with other methods you chose in the paper (CNN or transformer). But the results show that MLP / InversionMLP performs best in most cases. I am concerned whether all methods are well-trained and under the best hyperparameters.

**Questions:**

- If you are sure all methods are well-trained and under the best hyperparameters, could you explain why MLPs perform better than methods like CNN and Transformer in 3D cases, which seems counterintuitive? Or, could you provide additional analysis or ablation studies that might shed light on why MLPs are outperforming other architectures?

- Could you compare the performance of these methods in scenarios of out-of-distribution?

- In line 396, the paper says “To evaluate the flexibility of FWI”. Why is this  FWI? I think this section is talking about forward modeling. Could you clarify whether this section is indeed about forward modeling rather than FWI?

---

> ### Author Response · Authors · 2024-11-14
> **Response to Reviewer 8xHL**
>
> We appreciate your recognition of the dataset's potential benefit to the community, as well as your positive assesment on our presentation.
>
> > I think the machine learning models are kind of simple. Based on my experience, MLPs seem not to be the best choices for 3D problems compared with other methods you chose in the paper (CNN or transformer). But the results show that MLP / InversionMLP performs best in most cases. I am concerned whether all methods are well-trained and under the best hyperparameters.
>
> We have try our best to optimize our models using hyperparamter search techniques. While MLP is a fundamental machine learning model, its strong performance in our study aligns with its ability to generalize across broader scientific domains. A similar observation was made in the Climsim dataset for climate prediction [1], the best paper in NeurIPS Dataset and Benchmark Track 2024, where MLP also outperformed more complex models like CNN, RPN, and cVAE. As we noted in Section 3.2.1, MLP's advantage lies in predicting the output in a vector space rather than a point space, as used in methods like DeepONet and GNOT. Thus, the model parameters of MLPs will be 20 times more than those in the point-based models. These point-based models, despite their sophistication, may struggle with convergence on large-scale problems, even with 10 times the training steps required by MLP (see Appendix E.3 for training details). Additionally, given the sparse spatial-temporal structure of wavefield and seismogram data, localized feature extraction—which CNNs excel at—is less critical in this case. MLP, with its fully connected layers, effectively captures global patterns in our seismic data even when the signals are sparse.
>
> [1] Yu S, Hannah W, Peng L, et al. ClimSim: A large multi-scale dataset for hybrid physics-ML climate emulation. Advances in Neural Information Processing Systems, 2024.
>
> > Could you compare the performance of these methods in scenarios of out-of-distribution?
>
> Thank you for this valuable suggestion regarding out-of-distribution (OOD) generalizability. We acknowledge the importance of exploring OOD cases in ML studies; however, given the uniqueness of Earth as our subject, creating OOD scenarios falls outside our primary objective of achieving accurate Earth tomography. Our design relies on well-established geological studies [1-3] that provide a credible distribution of Earth's internal properties, ensuring our model is grounded in recognized physical and structural characteristics. Expanding to OOD cases would involve hypothetical or artificially generated distributions, potentially misaligned with realistic geological relevance.
>
> [1]  Dziewonski A M, Woodhouse J H. Global images of the Earth's interior. Science, 1987.
>
> [2] French S W, Romanowicz B A. Whole-mantle radially anisotropic shear velocity structure from spectral-element waveform tomography. Geophysical Journal International, 2014.
>
> [3] Tromp J. Seismic wavefield imaging of Earth’s interior across scales. Nature Reviews Earth & Environment, 2020.
>
> > In line 396, the paper says “To evaluate the flexibility of FWI”. Why is this FWI? I think this section is talking about forward modeling. Could you clarify whether this section is indeed about forward modeling rather than FWI?
>
> Thank you for your reminder. This section is focused on forward modeling at higher resolution. The reference to FWI is intended to highlight the significance of conducting higher-resolution forward modeling, which is to achieve better inversion results in FWI across different resolution scales. We will change the sentence to "To evaluate seismic modeling across different resolution scales".

---

> > ### Comment · Reviewer_8xHL · 2024-11-26
> >
> > Thank you for the responses. I basically agree with your explanation about MLP’s good performance, although I still believe there are some other neural networks more suitable for these datasets. I retain my current score at 6.

---

> > > ### Author Response · Authors · 2024-11-26
> > >
> > > Thank you for your comments. We would appreciate it if you could provide us with some specific alternatives that are more suitable for our datasets.

---

> > > ### Author Response · Authors · 2024-12-01
> > >
> > > Dear Reviewer 8xHL,
> > >
> > > We appreciate your feedback and would like to clarify that we have carefully considered a range of networks to ensure the best fit for our dataset.
> > >
> > > 1. **MLP**: As a classic baseline, MLPs excel at capturing **global patterns** in seismic data, which is crucial for Earth tomography. Their fully connected layers enable effective generalization, especially for sparse datasets, as demonstrated in studies like **ClimSim**, where MLPs outperformed more complex models.
> > >
> > > 2. **Fourier Networks**: These networks are particularly effective for capturing **frequency-domain features** in seismic data, which are essential for accurate wavefield analysis and seismic inversion.
> > >
> > > 3. **DeepONet**: Designed for **operator learning**, DeepONet is ideal for mapping complex relationships between seismic inputs and outputs, enabling better handling of non-linear data in PDE-based problems.
> > >
> > > 4. **GNOT**: GNOTs excel in modeling **spatial-temporal dependencies** using the transformer architecture, which is critical for seismic data that exhibit complex correlations across both space and time.
> > >
> > > 5. **Physics-informed DeepONet**: This model combines operator learning with **physics-based constraints**, enabling it to model non-linear relationships in higher resolution while ensuring adherence to physical laws.
> > >
> > > 6. **CNN**: While CNNs excel at **local feature extraction**, their role in our study is secondary to MLPs in handling seismogram data. However, they are still valuable for capturing localized spatial patterns in seismic data when needed.
> > >
> > > Although we also considered other neural operators like FNO and its variants, they were not suitable for our dataset due to their assumption of a **regular mesh**, which does not align with the global nature of our data.
> > >
> > > We hope this explanation clarifies that we have thoroughly evaluated the suitability of possible networks for our dataset, taking into account key factors to achieve optimal modeling.

---

### Official Review · Reviewer_g4Tn · 2024-11-02

**Soundness:** 2
**Presentation:** 3
**Contribution:** 2
**Rating:** 6
**Confidence:** 5

**Summary:**

The paper, "GlobalTomo: A Global Dataset for Physics-ML Seismic Wavefield Modeling and FWI," introduces a 3D global synthetic dataset designed specifically for machine learning applications in seismic wavefield modeling and full-waveform inversion. The GlobalTomo dataset integrates high-resolution seismic simulations that span various scales, from acoustic and elastic wave propagation to planetary-scale simulations representing real Earth conditions. The paper demonstrates the utility of this dataset by benchmarking several machine learning models for both forward modeling and inversion tasks, underscoring the potential of ML to accelerate seismic data interpretation and enhance our understanding of Earth’s interior.

**Strengths:**

This paper addresses a key gap by providing a global-scale dataset that integrates ML-friendly features with robust physical modeling for seismic applications. The dataset’s design across three tiers (acoustic, elastic, and real Earth) allows it to support scalable and complex seismic modeling tasks.

**Weaknesses:**

1. Wavefield Estimation. The necessity of including the wavefield as part of the dataset is not entirely convincing. While the authors provide further discussion on this in Supplementary Section F.2, this inclusion could be misleading for two primary reasons.

First, from a practical inverse problem perspective, only surface wavefield measurements (seismograms) are typically available in real-world FWI applications. The full wavefield throughout the Earth’s interior is generally inaccessible and, therefore, impractical for use in inversion.

Second, traditional model-based FWI methods do require calculating the full wavefield as they rely on directly solving the wave equation. However, end-to-end ML-based inversion strategies approximate the inversion process and thus do not depend on the full wavefield. These methods instead focus on estimating target parameters directly from surface measurements, bypassing the need for a complete wavefield solution.


2. Comprehensive Assessment of the Dataset.  An essential purpose of a benchmark dataset is to support various realistic scenarios that may arise in practical applications. A key example would be testing generalization capabilities. However, this aspect is not clearly addressed in the manuscript.

It remains unclear whether the dataset, or its three tiers, adequately represents different distributions or if they cover a range of scenarios that would facilitate generalization. No visualizations are provided to illustrate sample distributions, making it difficult to assess the dataset’s diversity. Additionally, I did not find any generalization tests conducted with direct inversion methods, as outlined in Section 3.2.2. This raises concerns about the dataset's robustness across diverse inversion tasks.

**Questions:**

1. Could the authors clarify the intended benefit of including the full wavefield in the dataset, given that only surface measurements are typically available in realistic FWI scenarios? How does the inclusion of the full wavefield impact the dataset’s applicability to practical, real-world inversion problems where the wavefield is not accessible?


2. Could the authors provide more details on the distribution of samples within each tier of the dataset? Are there visualizations available to illustrate the diversity of scenarios represented?


3. Were any generalization tests conducted using direct inversion methods to assess the dataset’s performance across varied inversion scenarios? If not, it would be much appreciated if some generation tests could be conducted using the direction inversion methods used in this current manuscripts (InversionNet-3D, InversionMLP).

---

> ### Author Response · Authors · 2024-11-14
> **Response to Reviewer g4Tn**
>
> We appreciate your acknowledgment of our contributions in addressing key gaps in seismic data modeling and inversion, as well as your constructive feedback on areas for further improvement.
>
> > Wavefield Estimation. Question 1.
>
> We appreciate the reviewer’s question regarding the inclusion of the full wavefield in the dataset. The primary motivation for incorporating the full wavefield is twofold:
> 1. **Physics-informed training**: Including the full wavefield allows the training process to incorporate rich, multi-scale physical constraints (obey the wave equation) that could regularize the inversion outcomes. By leveraging the complete spatiotemporal data, the model can learn physics-informed patterns in wave propagation throughout the subsurface layers. In contrast, purely fitting the subsurface data may fail to ensure physical consistency.
> 2. **Interpretability and scientific insight**: Access to the complete wavefield enables us to interpret model outputs in relation to the underlying geophysical processes, enhancing the scientific interpretability of the learned representations, and avoidng potential black-box mapping and data overfitting. Training with the full wavefield allows for step-by-step insights into how subsurface structures influence wave signals, which ultimately manifest in surface seismograms. This interpretability is crucial for geological researchers to assess the reliability of inversion outputs, particularly when only surface data is available during actual deployment.
>
> > It remains unclear whether the dataset, or its three tiers, adequately represents different distributions or if they cover a range of scenarios that would facilitate generalization. Question 2.
>
> The dataset’s design is informed by geological studies [1-3] that accurately reflect Earth's internal structural parameters, specifically through the use of spherical harmonics. These harmonics, sampled using Latin Hypercube Sampling from -10% to 10% (Line 191), ensure comprehensive coverage across the entire design space of interest, thereby grounding the model in well-recognized physical and structural distributions . As our goal is to inverse our unique Earth's structure, we didn't consider out-of-distribution (OOD) generalization since OOD cases would involve hypothetical or artificially generated distributions, potentially misaligned with realistic geological relevance. Regarding the diversity of the three tiers, they incorporate a range of physical processes and equations that capture the dynamic difference of seismic scenarios, thereby enhancing our dataset’s applicability in various seismology applications. We will provide improved illustrations in the main text and table 1 to better showcase the dataset’s diversity.
>
> [1]  Dziewonski A M, Woodhouse J H. Global images of the Earth's interior. Science, 1987.
>
> [2] French S W, Romanowicz B A. Whole-mantle radially anisotropic shear velocity structure from spectral-element waveform tomography. Geophysical Journal International, 2014.
>
> [3] Tromp J. Seismic wavefield imaging of Earth’s interior across scales. Nature Reviews Earth & Environment, 2020.
>
> > Additionally, I did not find any generalization tests conducted with direct inversion methods, as outlined in Section 3.2.2. Question 3.
>
> There seems to be some misunderstanding. The results for direct inversion methods in Section 3.2.2 are indeed evaluated on an unseen test set, using the same 90% training and 10% test split as our other experiments to assess generalization. We will make this clearer in the revised manuscript.

---

> > ### Comment · Reviewer_g4Tn · 2024-11-23
> >
> > I appreciate your responses and have decided to retain my current score.

---

### Meta-Review · Area_Chair_z4fV · 2024-12-22

**Metareview:**

The authors introduce the GlobalTomo dataset, which is a large 3D synthetic dataset tailored for seismic wavefield modeling and full-waveform tomography. For this problem there already exist a handful of publicly available datasets (such as OpenFWI and OpenFWI 2.0), The main novelty appears to be that while the previous datasets have focused on subsurface reconstruction and imaging, the proposed dataset includes "global-scale" wavefields (capturing core-to-surface phenomena at the planet scale).

There is considerable computational effort involved in generating this dataset, but the authors do a good job of describing all the necessary details. Overall, the paper is well-written, and reviewers pointed out that the dataset can be valuable to the community.

However, some concerns remained, particularly related to bias and generalization. There is no ground truth for the earth's interior; all existing models are inaccurate, and bias is inevitable in synthetically generated datasets; and performance may not generalize. Moreover, benchmark numbers on existing approaches may favor certain types of approaches over other ones. For these reasons I think the numerical results (and the rank ordering of the different inversion methods) reported in this manuscript need to be taken with appropriate caveats.

This was a tough decision, but in the end the concern of generalizability is serious enough (and certainly not out of scope, as the authors argued in their response). The issue of sensitivity to OOD models warrants a more thorough discussion in the main paper with additional numerical results, over and above the checkerboard tests reported in the appendix. For maximizing impact, please consider using the best practices already established in the FWI community while constructing these additional studies.

**Additional Comments On Reviewer Discussion:**

There was a healthy back-and-forth between the reviewer. Initial scores were low/borderline

The main issue raised by a reviewer (who is an expert in the area) is that of bias in the dataset, and the related issue of out-of-distribution performance. Another reviewer also raised the issue of counter-intuitive performance of MLPs vs operator networks , but this, in my view, not a critical concern. In response to the reviews, the authors argued that OOD generalization studies are out-of-scope (since the earth is unique) but this doesn't quite suffice in my opinion.

---

> ### Public Comment · ~Chaoshun_Hu1 · 2025-10-27
> **Suggested Views from a domain expert**
>
> In geophysics, global geophysics and exploration or controlled geophysics are two different branches. These authors published a global seismic data which aims at large scale or global scales. Open FWI are aimed at exploration especially in oil and gas. I think this global dataset is very important for earthquake prediction, natural hazards.

---

### Decision · Program_Chairs · 2025-01-22

Reject